# A systematic approach to orient the human protein–protein interaction network

Dana Silverbush[1] & Roded Sharan[1]

The protein-protein interaction (PPI) network of an organism serves as a skeleton for its signaling circuitry, which mediates cellular response to environmental and genetic cues. Understanding this circuitry could improve the prediction of gene function and cellular behavior in response to diverse signals. To realize this potential, one has to comprehensively map PPIs and their directions of signal flow. While the quality and the volume of identified human PPIs improved dramatically over the last decade, the directions of these interactions are still mostly unknown, thus precluding subsequent prediction and modeling efforts. Here we present a systematic approach to orient the human PPI network using drug response and cancer genomic data. We provide a diffusion-based method for the orientation task that significantly outperforms existing methods. The oriented network leads to improved prioritization of cancer driver genes and drug targets compared to the state-of-the-art unoriented network.

[1] The Blavatnik School of Computer Science, Tel Aviv University, Tel Aviv 69978, Israel. Correspondence and requests for materials should be addressed to R.S. (email: roded@tau.ac.il)

High-throughput technologies are routinely used nowadays for interactome mapping by technologies such as yeast two-hybrid[1,2] and co-immunoprecipitation followed by mass spectrometry[3]. However, the resulting protein–protein interaction (PPI) networks have limited predictive power as they lack information on the logic of the connections. Going beyond bare topological models requires, as a first step, orienting the interactome, that is, predicting the direction of signal flow (if any) of each interaction[4]. Such orientation information can be further utilized to predict common properties of signaling pathways[5], study the progression of genome-related diseases[6], aid in the development of drugs[7] and tailoring treatment combinations[8], investigate the effects of a chemical inhibitor in a disease setting[9], and many more. Indeed, Cao et al.[10] showed that incorporating the direction of interactions improves the prediction of protein function in yeast. In human, Vinayagam et al.[11] showed that a directed network improves the prediction of previously unknown modulators in the ERK signaling pathway. Nevertheless, to date, direction information is available for only a small percentage of the interactions. For example, Kyoto Encyclopedia of Genes and Genomes (KEGG)[12] contains 5769 directed interactions in human out of the current 311,962 interactions present in Bio-GRID interaction database (BioGRID)[13], where we expect at least 40% of these interactions to have a direction[14].

Existing orientation methods have been mainly applied to yeast[15–19]. These approaches relied on information from perturbation experiments, in which a gene is perturbed (cause) and as a result other genes change their expression levels (effects). The common assumption in these methods is that for an effect to take place there must be a directed path in the network from the causal gene to the affected gene.

The generalization of these methods to human is hampered by two main obstacles. First, there is a lack of cause-effect information to guide the orientation. Second, the scale of the problem is much bigger as the size of the human PPI network is almost 3-fold bigger than the yeast one (BioGRID, January 2016, non-redundant Interactions)[13]. The only attempt to orient a human PPI network was made by Vinayagam et al.[11]. They overcame the lack of a comprehensive collection of cause-effect pairs by focusing on (shortest) paths from membrane receptors to transcription factors (TFs). As the correspondence between receptors and TFs is only partially known, they assumed that all receptors should be connected to all TFs, thus obtaining only an approximation to the true signaling directions.

Here we report on a method for network orientation, Diffuse2Direct, which is based on diffusing signals from causal proteins to affected proteins. To generate cause-effect information in human we utilize two independent resources: (i) drug response data, which captures the effect of drugs, represented by their protein targets, on gene expression; and (ii) cancer genomic data, which captures the effect of a patient's somatic mutations on gene expression. We show that the assigned directions are highly accurate, outperforming the state-of-the-art methods by a wide margin. Moreover, we show that the predicted directions are robustly predicted and are consistent across the data source used. We then integrate all available data sources to construct a consensus-oriented human network. We demonstrate its utility in the prediction of drug targets and cancer driver genes.

Diffuse2Direct is available open source on github at: https://github.com/danasilv/Diffuse2Direct

## Results

**Diffuse2Direct: diffusion-based approach to orient a network.** Diffuse2Direct (D2D) uses causal proteins and paired effects to orient an undirected or a partially directed network (Fig. 1). The input to D2D consists of a collection of experiments, also termed guiding sources, each of which induces a set of causal proteins and a set of affected proteins (Fig. 1a). Here, we have considered drug response data, where causal proteins correspond to drug targets and affected proteins correspond to genes whose expression changed following a treatment with the drug. Similarly, we have exploited cancer genomics data, where causal proteins correspond to mutated genes and affected proteins correspond to genes that are differentially expressed between the tumor and matched-normal tissues. The input to D2D further includes a physical network of protein–protein and protein–DNA interactions (Fig. 1a). Each experiment is used to compute a network diffusion value for each protein in the network according to its proximity to the causal proteins of that experiment on the one hand, and to the affected set on the other hand (Fig. 1b). These diffusion values are combined to score the likelihood of each of the two possible directions of an edge, and the resulting scores are used as predictive features for the edge's true direction (Fig. 1c). A classifier is then applied to the computed features to predict the direction of the interactions (Fig. 1d). Each interaction is assigned with the most likely direction and a confidence estimate for it. For any given confidence threshold, the result is a partially directed network whose directed part (oriented interactions) can be viewed as representing signaling interactions and its undirected part (unoriented interactions) can be viewed as representing intra-complex or other undirected interactions.

To exemplify the predictive power of the edge scores computed via diffusion, we calculated them in a small scale example—the VEGF pathway from KEGG—stripping it off its orientation information. In this application, the network nodes were set to be the pathway's entities, and the edges were defined to be the pathway's interactions (in their undirected form). A score was computed for each direction of an edge by calculating the ratio of (1) diffusing from the pathway's set of input nodes (in-degree 0), and (2) diffusing from the pathway's set of output nodes (out-degree 0) as described in the "Methods" section. Since in this example there is only one guiding source (the inputs and outputs of the pathway), no subsequent classification step was needed and we can view the computed scores as the final D2D scores. The resulting scores were used to orient the pathway: if one direction received a higher score than its opposite direction, then this direction is chosen; otherwise, the edge is left undirected. The output network is depicted in Fig. 2a. Out of 52 interactions, 48 are oriented as cataloged in KEGG (92% accuracy), including the complex FAK-Paxillim whose interaction remains undirected. In comparison, a naïve orientation strategy based on depth first search (DFS) or breadth first search (BFS) visiting time (i.e., orienting an edge from the first visited to the second visited node) yielded inferior results with accuracies of 57% and 81%, respectively. The higher performance of the diffusion-based strategy, even in this simple setting, can be explained by the consideration of multiple paths at the same time, rather than a single path at a time.

A similar high concordance between the predicted and true orientations in KEGG was obtained when analyzing the five largest pathways in KEGG (containing 200–350 irreversible interactions each), with 88% agreement on average, as well as when analyzing directed interactions from NetPath[20], downloaded via PathLinker[21] (Fig. 2b). We show-case the scores distribution for the different KEGG interaction types via the pathway *Steroid hormone biosynthesis*, showing a clear separation between the scores of oriented interactions in KEGG and the scores of interactions that should remain unoriented (Fig. 2c). For the latter, D2D scores yielded a distribution centered around 0, capturing the lack of directionality for these interactions.

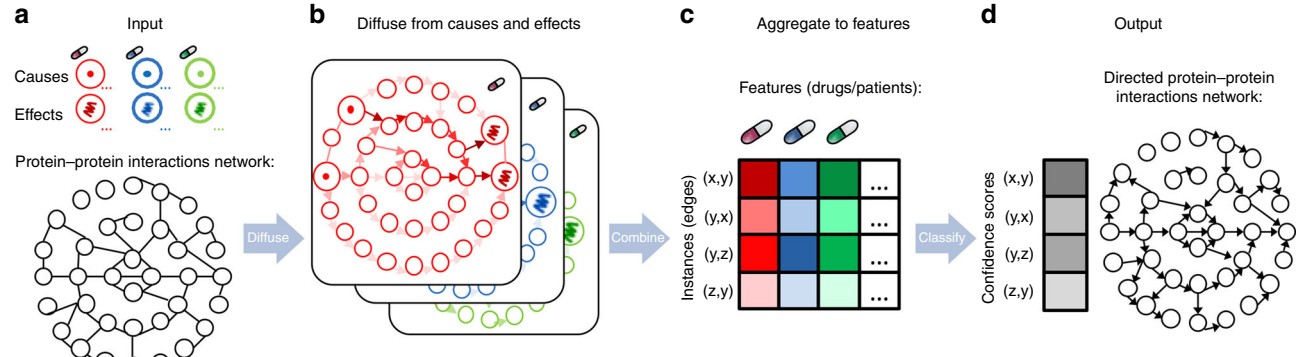

**Fig. 1** An overview of the Diffuse2Direct orientation algorithm. **a** We start from an input network along with guiding information consisting of a collection of experiments, each of which induces a set of causal proteins and a set of effect proteins. As an example we show here drug response information, inducing a set of drug targets and their associated set of differentially expressed genes. **b** For each experiment in the collection, we perform network diffusion once from the causal set and once from the effect set, and integrate the scores. **c** The resulting scores are used as predictive features for the direction of each edge. **d** A classifier assigns a confidence to each directed edge, and using a cutoff we obtain a directed network

**Application to drug response data and performance evaluation.** As a first large scale application of the method, we used drug response data (see "Methods" section) to orient a human PPI network. Specifically, in this application we assumed that the effect of each drug can be described as emanating from its set of targets and affecting the genes that were observed to be differentially expressed in response to treatment with the drug. As test sets for the orientation we used five subsets of directed interactions (Supplementary Table 1): interactions from large scale databases, including kinases and phosphatases to their substrates (KPIs), protein–DNA interactions (PDIs) and E3 ubiquitination (E3) interactions—all down sampled to avoid degree bias (see "Methods" section). In addition, we used two sources of small-scale experiments: directed interactions from the well studied EGFR pathway (EGFR)[22], and a collection of signal-transduction interactions in mammalians compiled by Vinayagam et al.[11] (STKE).

To evaluate the predicted orientation we computed precision-recall curves for each of the five test sets under a 3-fold cross-validation scheme (Fig. 3). Areas under these curves are given in Supplementary Table 1, ranging from 0.74 to 0.92 depending on the test data used. To compare our results to those obtained using the network topology alone, we applied a variant of the diffusion suggested by Erten et al.[23], which does not depend on prior cause-effect information. This topology-only variant results in an average area under the curve of 0.56 (Supplementary Table 1), demonstrating the importance of the causality information in guiding the orientation. Robustness analysis shows that the high performances are consistent across parameter choice (Supplementary Fig. 1).

We compared D2D to two previous orientation methods. The first method by Vinayagam et al.[11] tackles the lack of experimental guiding data by using the functional annotation of proteins. Vinayagam et al. computed features describing the probability of each direction of an interaction to participate in a shortest path from any membrane receptor to any transcription factor. The second approach, called SHORTEST[17], is a leading approach for orienting the yeast network, guided by knockout experiments in which one gene is perturbed (cause), and as a result other genes change their expression (effects). Using integer linear programming (ILP) formulation SHORTEST orients the network so as to connect a maximum number of cause-effect pairs via shortest paths. We adapted SHORTEST to orient the human network by using the drug response data and by modifying the confidence assignment scheme of SHORTEST to

make it scalable to the human network (see "Methods" section). Notably, the fraction of edges assigned with a direction by SHORTEST depends on the edge coverage by the shortest paths connecting the input pairs. For the drug response data, such paths cover only 44,647 of the edges, thus greatly limiting the method's coverage. For comparison purpose, when including the SHORTEST orientation we restricted the evaluation to the interactions covered by SHORTEST (Supplementary Fig. 2). Vinayagam et al. outperforms SHORTEST on the restricted set, while D2D outperforms both. On the full network, D2D outperforms Vinayagam et al. on all test sets (Fig. 3).

To evaluate the effect of the size of the cause-effect input data, we used growing numbers of drugs to orient the network. As evident from Supplementary Fig. 3, increasing the number of drugs increases both the recall and the precision of the predictions, supporting our use of drug information for the orientation.

Next, we aimed to evaluate the contribution of different drug attributes to the orientation. To this end, we extracted from the classifier the number of times each drug was chosen as a feature (i.e., received a non-zero coefficient in the L1 regularization setting) and compared it to: (i) the number of its known drug targets, (ii) the number of genes that were observed to be differentially expressed in response to treatment with the drug, (iii) the cellular localization[24] of its targets, and iv) the cellular localizations of the genes that were observed to be differentially expressed in response to the drug (Supplementary Fig. 4). The attribute that was most correlated with choosing the drug as a feature was its number of known targets that are localized to the membrane (Pearson correlation two-sided 0.2, $p$-value $< 10^{-6}$), with the next being number of targets localized to the extracellular matrix (Pearson correlation two-sided 0.12, $p$-value $< 0.0025$) and the third being the number of known targets for a drug (Pearson correlation two-sided 0.1, $p$-value $< 0.0088$). Although statistically insignificant, it is interesting to note that the fourth attribute was the number of differentially expressed genes that are localized to the nucleus (Pearson correlation two-sided 0.06, $p$-value $< 0.1245$).

**Application to cancer genomics data.** As an alternative to the drug-based pairs, we used gene pairs derived from genomic mutation data and the resulting expression changes from The Cancer Genome Atlas (TCGA). In this setting, mutations, fusions and copy number alterations of a patient are translated to causal

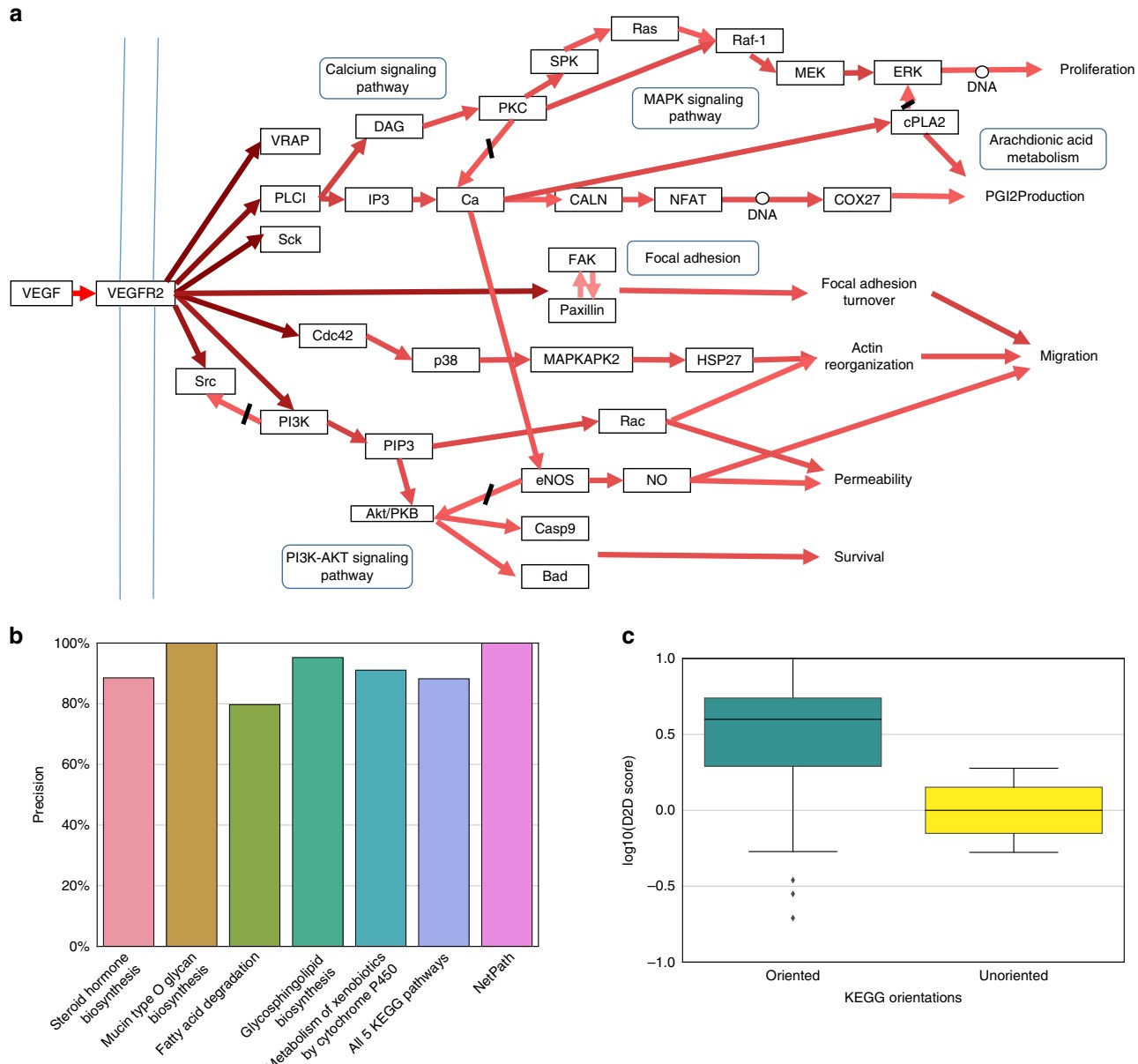

**Fig. 2** Small-scale examples of D2D on KEGG pathways. **a** The *VEGF signaling pathway* from KEGG is oriented based on D2D scores from a single experiment, connecting input nodes (VEGF) to output nodes (VRAP, Sck, Proliferation, PGI2Production, Migration, Permeability and Survival). The figure depicts the directions chosen for each of the interactions, where the color of an interaction represents the confidence in its orientation, the direction of an arrow indicates the inferred interaction direction, and a black mark indicates a direction inference which is inconsistent with the KEGG annotation. 92% of the interactions are oriented as cataloged in KEGG. **b** Similar high concordance between the predicted and true orientations in KEGG is obtained when analyzing the five largest pathways in KEGG and the NetPath resource. **c** Show-case of D2D score distribution in the *Steroid hormone biosynthesis pathway* from KEGG, separated to the scores assigned to the oriented interactions in KEGG and the scores assigned to those interactions that are unoriented in KEGG. The box plots were plotted using seaborn python package with the default setting such that the center line stands for the mean of the distribution, the box extends from the first to the third quartile of the distribution and the whiskers extend to 1.5 IQR

proteins, and genes that are differentially expressed in a tumor with respect to a matched healthy tissue serve as the effects (see "Methods" section). We oriented the human network using guiding information from four sources: 200 acute myeloid leukemia (AML)[25], 960 breast cancer[26], 631 colon cancer[27], and 316 ovarian cancer[28] patient samples. To evaluate the results we used the same cross-validation scheme as above (Supplementary Table 1). Remarkably, we observed wide agreement on the orientation derived from each of the guiding sources (Fig. 4a). Furthermore, the higher the agreement the higher is the orientation accuracy, with 18.2% accuracy for directions supported by only one

oriented network, 39.6% for directions supported by two sources, 60.4% for three, 81.8% for four, and 96.3% for directions supported by all five oriented networks (Fig. 4b, chi-square $p < 10^{-40}$).

**Construction and evaluation of a consensus orientation.** Reassured by the good performance and high agreement on the test sets, we turned to learn an orientation by using all available training data (i.e., using the five test sets amounting to 33,756 directed interactions) and integrating all available guiding sources. To this end, we used a single guiding source at a time (drug

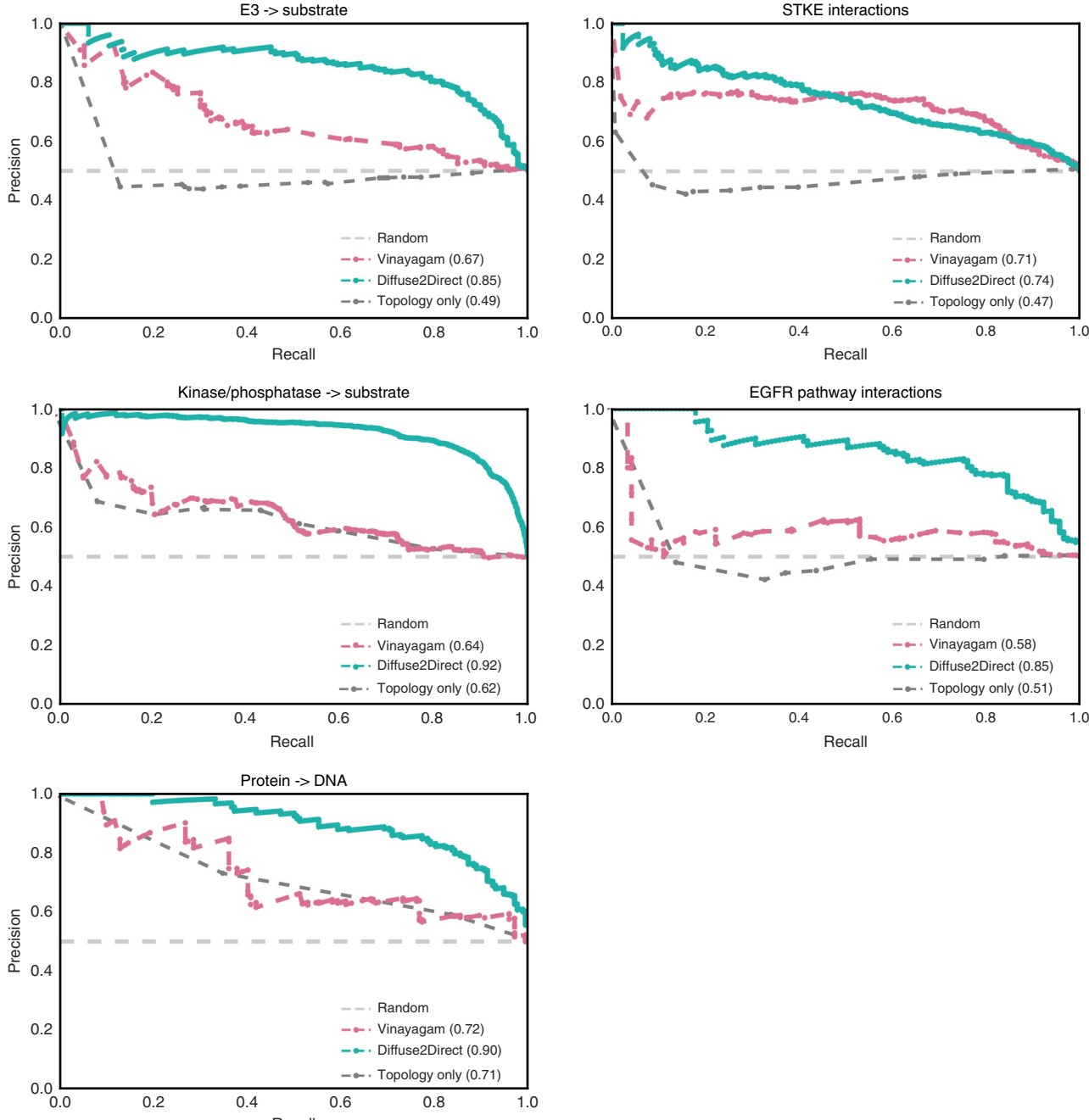

**Fig. 3** Performance evaluation. Shown are cross-validation results comparing the directions inferred by Diffuse2Direct, Vinayagam et al. and a benchmark topology-based approach to subsets of interactions with known directionality: E3 ubiquitination interactions, kinases to their substrates (KPIs), protein–DNA interactions (PDIs), signal-transduction interactions in mammalians (STKE) and interactions from the well-studied EGFR pathway. The area under the precision-recall curve for each approach appears in parentheses

response, AML, breast, colon or ovarian patients) training a classifier with the directed 33,756 interactions so as to assign direction estimates to the remaining 145,631 undirected interactions. Whenever the ratio D2D($u,v$) to D2D($v,u$) exceeded $1 + \in$ (where we set $\in$ to be 0.01) for some information source, we included the edge ($u,v$) in its directed network (Supplementary Table 2). Next, we constructed a consensus-oriented network by counting the votes for each direction from the five source-specific directed networks. The resulting oriented network is available as Supplementary Data 1 and includes for each edge its orientation confidence, computed as the number of guiding sources supporting the inferred orientation divided by the number of guiding

sources supporting the opposite orientation (or by 1 if there is no support for the opposite orientation). We marked the edges with low orientation confidence (less than 2) as unoriented, and the rest as oriented, leaving 69% of the network's edges oriented. Validating this network with independent sources suggests that it is highly reliable. Out of 33% (69,704) of the network's edges that were oriented with maximum confidence, 1917 were included in curated information from the PathLinker database[21] with 70% agreement (1334 had the same orientation and 583 were oriented differently, hyper geometric $p$-value $< 10^{-90}$). Out of 69% of the edges that were oriented with confidence of at least 2 confidence score, 2,815 were included in PathLinker with 63% agreement

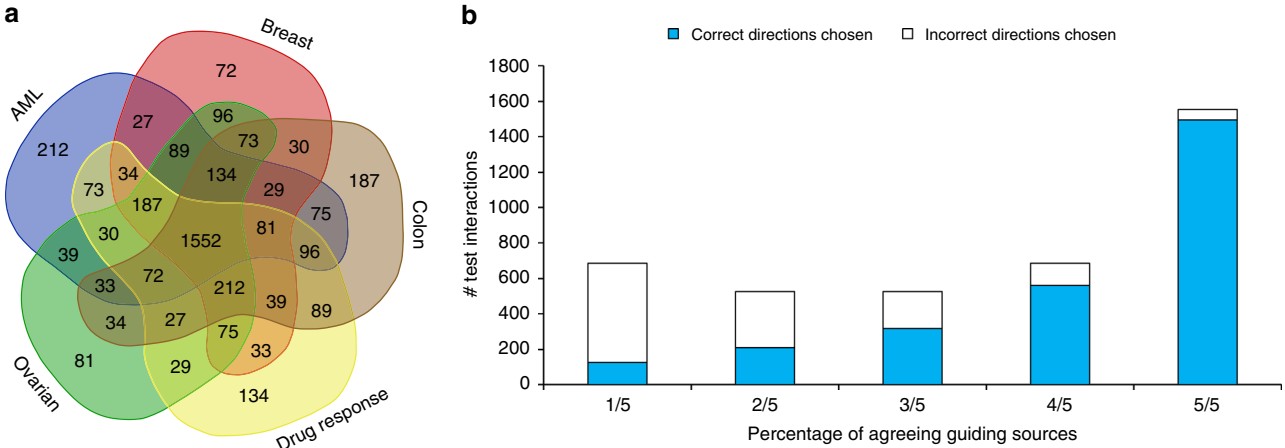

**Fig. 4** Concurrence of orientations according to different sources of cause-effect information. **a** Depicted are the agreement of directions assigned to the test edges when independently orienting the network using (i) drug targets to differentially expressed genes, or (ii) genetic perturbations to differentially expressed genes across different cancer patients (AML, breast, colon and ovarian cancer). **b** The higher the agreement the higher is the orientation accuracy, which goes up to 96% when all sources agree

($p < 2 \times 10^{-67}$). As another validation, the interactions that D2D left as unoriented were enriched with known complexes from the CORUM database[29] with a hyper geometric $p$-value $< 10^{-102}$. When applying the same test to the highly confident oriented interactions (confidence of 5), we obtained an insignificant hyper geometric p-value of 0.99.

We reasoned that the resulting orientation can assist in elucidating the functional roles of proteins based on their network location, as the edge directions greatly limit the number of paths in the network. As an example, we applied the orientation to prioritize drug targets. We oriented the network using only the cancer data sets. Then for each drug, we ranked the genes as potential targets of the drug based on their proximity to the drug-induced expression. To this end, we flipped all directions in the (oriented) network, so that a diffusion process will follow a signal traversing up-wards in the network, i.e. opposite of the orientation, and computed a network diffusion score using the differentially expressed genes induced by the drug as a prior set. We found that the orientation-based scores outperform the ones obtained when using an unoriented network (Fig. 5a), with a mean rank for a true drug target of 6045 (out of 15,501 network genes) compared to a mean rank of 8837 obtained by a completely unoriented network (Wilcoxon signed-rank test, $p$-value $< 10^{-12}$).

Similarly, we hypothesized that the oriented network could assist in elucidating cancer driver genes. We oriented the network leaving out one disease set at a time (AML, colon, breast or ovarian cancer). Then, we ranked the genes based on their proximity to the differentially expressed genes of the data set: as before we flipped the network, and computed the network diffusion scores for each patient separately, diffusing from the patient differentially expressed genes. We aggregated the ranking from all the patients to create one comprehensive list of putative driver genes. Focusing on the top K% scoring genes at a time, we calculated their enrichment against multiple sources of driver (positive controls) and non-driver genes (negative controls). To this end, we used the gene lists introduced by Hofree et al.[30], complied from different sources: Cancer Gene Census version 73 (CGC)[31], the Atlas of Genetics and Cytogenetics in Oncology and Hematology (AGO)[32], UniprotKB[33], DISEASES[34] and MSigDB[35]. The oriented network based computation consistently reported higher fractions of driver genes and lower fractions of non-driver genes compared

to the unoriented one (Fig. 5b for the top 1% genes in AML, Supplementary Fig. 5 for the full range of K values and four disease sets). In particular, 49% of the top 1% of the genes reported by the orientation-based computation are in the AGO positive list, which consists of 1429 genes (hyper geometric $p$-value $< 10^{-37}$); 30% of the top 1% reported genes are in the CGC positive list, which consists of copy number variations, single nucleotide variants, somatic mutations and translocation of 531 genes (hyper geometric $p$-value $< 10^{-29}$); 39% of the top 1% genes are in the text mining list derived from the DISEASES database which consists 711 genes (hyper geometric $p$-value $< 10^{-40}$), and 63% of the 1% genes found in the comprehensive list unifying CGC, UniprotKB, DISEASES and MSigDB[30] which consists of 2045 genes (hyper geometric $p$-value $< 10^{-47}$). Importantly, known non-driver genes are underrepresented in the predictions, with only 1.3% of the top 1% reported genes present in the curated negative AGO list (NegAGOClean) which consists of 3272 known non-driver genes (hyper geometric $p$-value $< 10^{-13}$).

## Discussion

We have developed a method for orienting a network that is based on diffusion. We applied it to multiple drug response and cancer genomics data sets to infer a comprehensive and highly accurate orientation of the human protein–protein interaction network, significantly outperforming previous work. Key to the power of the oriented network is the great reduction in the number of possible paths in the network, guiding any subsequent analysis to the more plausible ones. To exemplify the power of this reduction, we applied the oriented network to the inference of drug targets and cancer driver genes. In both tasks, it significantly outperformed an application that is based on the unoriented network.

The network we have constructed can be incorporated with minimum to no change into a variety of existing solutions: as a skeleton for inferring process-specific subnetworks[36–38], propagating functional and disease-related information[10,39–42], and learning logical models of signaling pathways with the potential to greatly expand our understanding of their inner workings[43–45]. Thus, we believe that the oriented network advances us an important step toward mechanistic understanding of cellular processes.

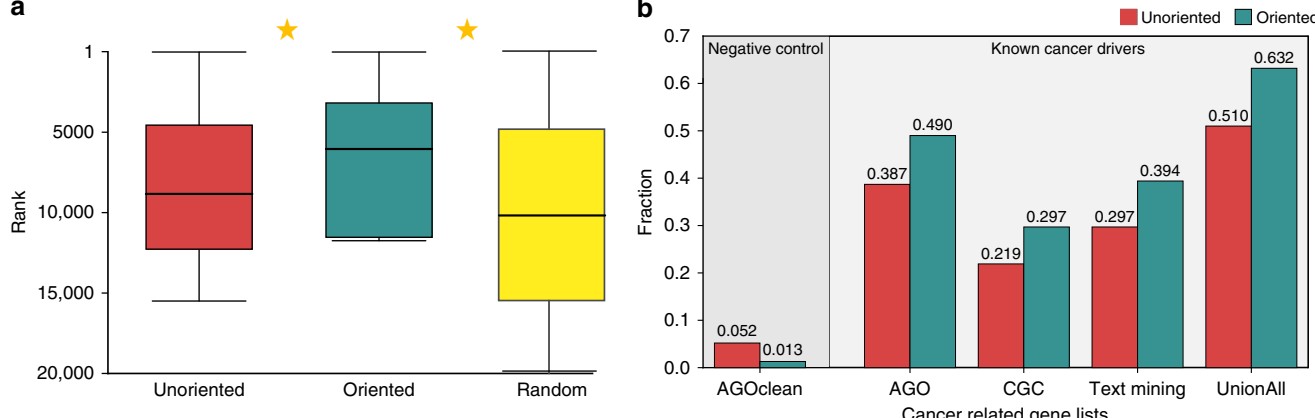

**Fig. 5** Orientation improves the prediction of drug targets and cancer driver genes. **a** The genes in the network are ranked according to their proximity to the differentially expressed genes of a given drug. We calculated the rank assigned to the correct drug targets by using the oriented network versus the unoriented one and a random ranking, and found that the oriented-based scoring ranks the correct drug targets higher (closer to 1) than both the unoriented-based scoring and the random scoring, with Kalgarov-Smirnov *p*-value < 0.05, indicated by a star. The box plots were plotted using seaborn python package with the default setting (as detailed before in Fig. 2c). **b** The genes in the network are ranked by their proximity to the differentially expressed genes of the cancer patients. Showing here the percentage of genes out of the top K = 1% ranked genes matching known driver genes and known non-driver from multiple sources for AML. The oriented network consistently reports more known driver genes and less known non-drover genes than the unoriented one, see extended results in Supplementary Fig. 5

## Methods

**Diffusion-based orientation**. We devised an orientation algorithm, Diffuse2Direct (D2D), which is based on propagating signals from causal to affected proteins and observing the direction of signal flow in the network. Unlike previous approaches, the algorithm does not rely on shortest paths only and accounts for all possible paths from source to target proteins. D2D receives as input a collection of experiments inducing paired sets of causes and effects, and a network of physical interactions between proteins for which an orientation is desired. The network may be undirected or mixed (containing both undirected interactions and directed ones). To ensure that the diffusion process converges, the network must be connected if undirected, and strongly connected if mixed (as is, e.g., the case for the BioGRID network we use here). Each edge (*i*, *j*) of the network has an associated confidence score *weight* (*i*, *j*), which may be equal to 1 for an unweighted network. D2D assigns directions to the edges using the following steps: (i) diffusing once from the causal proteins and once from the paired effect proteins; (ii) combining the two diffusion scores into a single score for each possible edge direction; and (iii) using the combined scores across multiple pairs as predictive features for inferring the direction of each interaction. These steps are described in detail below.

**The diffusion process and edge scoring**. The diffusion process computes a score for each protein which is the sum of a network term and a prior knowledge term. Formally, the score $F(v)$ of a node $v$ with a set of network neighbors $N(v)$ is:

$$F(v) = \alpha \left[ \sum_{u \in N(v)} F(u)w(u,v) \right] + (1-\alpha)P(v) \qquad (1)$$

where α is a smoothing parameter that balances between the network and the prior terms, $w$ is the edge weight normalized by the sum of outgoing edges[33]:

$$w(u,v) = \frac{weight(u,v)}{\sum_{k \in N(u)} weight(u,k)} \qquad (2)$$

and $P(v)$ is a prior score to a protein, which is set to $\frac{1}{|Priors|}$ if it is part of the prior set, and 0 otherwise. The diffusion score is computed in an iterative manner, as described by Cowen et al.[41], where at each iteration a vertex pushes its current score to its (out-going) neighbors, in proportion to the weight of the respective outgoing edge, until convergence. Convergence is achieved when the square root of the summed absolute changes in scores for the last iteration is below β, where we set β to be $10^{-5}$.

To assign a score for each edge direction, we compute a diffusion $F^c$ from the set of causes (i.e., the set of causal proteins induced by an experiment serves as the prior set in this network diffusion computation) and a diffusion $F^E$ from the set of effects, after reversing any pre-set directions in the network. If an edge (*u*,*v*) is directed from *u* to *v* then we expect *u* to be closer to the causal proteins than *v*, and *v* to be closer to the set of effects than *u*, thus the ratios $\frac{F^C(u)}{F^C(v)}$ and $\frac{F^E(v)}{F^E(u)}$ should be greater than 1. We take the score of the directed edge (*u*,*v*) to be the product of these ratios, i.e., the larger the score the more likely it is that the edge is directed

from *u* to *v* (Fig. 1b):

$$score(u,v) = \frac{F^C(u) \cdot F^E(v)}{F^C(v) \cdot F^E(u)} \qquad (3)$$

Note that this formulation tends to relax potential degree bias as each node is taken into consideration twice—once in the numerator and once in the denominator.

**Inference of directions**. The above process can be recorded in a vector in which each entry corresponds to a candidate directed edge in the network (for example, (*x*, *y*), (*y*, *x*), (*y*, *z*), (*z*, *y*) in Fig. 1c) and contains its score. The process may be repeated for multiple input experiments, for example the administration of different drugs, inducing paired sets of causes and effects, resulting in a matrix of edge scores, where rows correspond to edges and columns to experiments. Each row reflects the contribution of its directed edge to connect the corresponding paired sets of causes and effects induced by an experiment. These features are fed to a logistic regression classifier which assigns a D2D score to each directed edge (Fig. 1d). To direct an edge, we choose the highest scoring direction unless the ratio of both scores, $Max\{\frac{D2D(u,v)}{D2D(v,u)}, \frac{D2D(v,u)}{D2D(u,v)}\}$ is below $1 + \epsilon$, where we set $\epsilon$ to be 0.01. To avoid overfitting and restrict the number of features, we used L1 regularization where the value of the regularization balancing parameter was chosen via a nested 3-fold cross-validation in the range of $10^{-4}$ to $10^4$. For completeness, we provide a formal description of the classifier in Supplementary Note 1. We further describe other inference strategies that were considered and evaluated in Supplementary Note 2.

**Drug response data**. For each drug, we consider its known targets as causal proteins and genes whose expression changes in response to treatment by the drug as effects. Drug targets were extracted and assembled from DrugBank[46], DCDB[47], and KEGG DRUG[12] databases. Genes whose expression changed in response to the corresponding treatment were extracted from the Connectivity Map (CMap, build2[48]). CMap contains 6100 gene expression measurements in response to the administration of 1309 drugs and small molecules. These measurements were taken under different drug concentrations and on different cell-line types using the Affymetrix HG-U133A and HT-HG-U133A Array. In order to form drug-specific set of differentially expressed genes, we followed the normalization and filtering procedures described in Iskar et al.[49] (Supplementary Note 3). In total, we extracted 551 drugs with 1915 known targets (702 unique targets, 3.48 targets per drug on average). We extracted 21,424 differentially expressed genes (3487 unique genes, 38.88 genes per drug on average).

**Cancer genomics data**. We utilized data generated by the TCGA Research Network http://cancergenome.nih.gov/ downloaded on April 2016. Samples were taken from 200 AML patients, 631 Colon cancer patients, 960 breast cancer patients, and 316 ovarian cancer. Per sample, we labeled a gene as causal if it was either called as mutated or had copy number variation by TCGA. We labeled a gene as an effect if it was called as significantly differentially expressed by Cosmic[31], i.e., its absolute fold change *z*-score was above 2.

**Implementation of previous orientation methods**. We reimplemented the method of Vinayagam et al.[11]. To this end, we counted for each directed edge the number of shortest paths traversing it to connect (a) any membrane receptor to any transcription factor, and (b) any member of a known family of membrane receptors to any member of a known family of transcription factors. These counts were used to derive classification features for predicting edge directions as done by Vinayagam et al. In addition, we tried a variant of the method guided by drug response data, using pairs of drug-targets and responding (differentially expressed) genes, rather than membrane-to-transcription-factor pairs, yet only 16% of the interactions in the network were covered by shortest paths from drug response data, information which the method relies on.

A second method we compared to is the SHORTEST[17] method, which uses an integer linear programming algorithm to find an orientation that connects a maximum number of cause-effect pairs using shortest paths. To speed up its computation for large networks, we changed the scheme it applies to assign confidence to the edge orientations as follows. Let $S_{opt}$ be the value of an optimal solution computed by SHORTEST. To compute a measure of confidence in a given orientation of an edge $e = (v, w)$, SHORTEST originally reruns the ILP while forcing $e$ to carry the opposite orientation $(w, v)$ and sets its confidence value to $c = S_{opt} - S_e$, where $S_e$ is the maximum number of satisfied pairs for the modified instance. The repeated solution of many ILP instances is very costly. Instead, we extended SHORTEST to include weights on the pairs guiding the orientation. When all the weights are assigned 1, SHORTEST computes $S_{opt}$. To each weight we subsequently add random noise by sampling from a Gaussian distribution with mean 0 and variance 0.1. We repeat this process 1000 times and rerun the ILP. The final per-pair score assigned to a direction of an interaction is the fraction of times (out of 1000 repeats) it was oriented in that direction. We used this score as a first feature and the number of shortest paths traversing the edge as a second, feeding both to a logistic regression classifier which assigns a confidence score to each directed edge. We verified that this method is comparable to the previously used one on the yeast data set it was originally applied to (Supplementary Fig. 6).

Last, we applied a variant of diffusion approach suggested by Erten et al.[23] to compute a benchmark orientation which is based solely on the network topology. Specifically, we computed the diffusion scores by setting α in equation (1) to 1, thus eliminating the effect of the chosen priors. For each edge $(u,v)$ we calculated three such (topology-only) scores $F(u)$, $F(v)$ and $F(u)/F(v)$, and used them as features for a logistic regression classifier.

**Network and validation sets**. We used the Homo Sapiens network from BioGRID (release 3.4.126)[13] of 147,753 PPIs integrated with known collections of directed interactions with proteins in the BioGRID network:

(i) 450 signal-transduction interactions in mammalian cells (STKE) from the Database of Cell Signaling (http://stke.sciencemag.org/cm/ April 23, 2009 version) used for validation by Vinayagam et al.[11]

(ii) 117 interactions of the EGFR signaling pathway (EGFR) from Samaga et al.[22]

(iii) 4,293 kinase/phosphotase to substrate interactions (KPIs) from Phosphositeplus (www.phosphosite.org)[50].

(iv) 28,566 protein to DNA interactions (PDIs) downloaded from ChEA database: integrating genome-wide ChIP-X experiments[51].

(iiv) 330 E3 ubiquitination interactions, downloaded from hUbiquitome[52], a database of experimentally verified human ubiquitination enzymes and substrates.

Extending the BioGRID network with these sets resulted in an integrated network with 179,487 interactions (358,974 bi-directed interactions). Confidence score estimates for interactions are taken from the ANAT[53] software, which uses a logistic regression-based scheme based on the techniques by which an interaction was detected. Directed interactions added to the network are assigned a fixed confidence value of 0.8.

For validation purposes, we used the known collections of directed interactions. We note that although PDIs and KPIs are routinely used as test sets for network orientation, both are strongly biased toward having interactions directed from a high degree protein to a low degree protein. To remove this degree bias, we filtered the test sets by random downsampling of such interactions so that the resulting number of interactions from high to lower degree nodes is equal to the number of interactions that are directed from low to higher degree nodes. The filtered test sets included: (i) 450 STKE interactions; (ii) 117 interactions of the EGFR signaling pathway; (iii) 1798 KPIs down-sampled from an original set of 4293 interactions; (iv) 171 PDIs down-sampled from 28,566 interactions; and (v) 104 E3 ubiquitination interactions down-sampled from 330 interactions. Each test set of true orientations was supplemented with an equal-size set of false orientations by using the same edges but flipping their directions.

**Reporting summary**. Further information on research design is available in the Nature Research Reporting Summary linked to this article.

## Data availability

The data that support the findings of this study are available in public repositories: (1) skeleton unoriented network is available on BioGRID (release 3.4.126) at https://thebiogrid. org/; (2) drug targets are available at DrugBank (https://www.drugbank.ca/), DCDB (http://www.cls.zju.edu.cn/dcdb/), and KEGG DRUG (https://www.genome.jp/kegg/); (3) Genes whose expression changed in response to the corresponding treatment are available from the Connectivity Map (CMap, build2, https://portals.broadinstitute.org/cmap/); (4) cancer genomics data are available from the TCGA research center (http://cancergenome.nih.gov/). (5) The consensus oriented network resulted in this study is available here as Supplementary Data 1.

## Code availability

The software is available open source on github at https://github.com/danasilv/Diffuse2Direct.

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

## Acknowledgements

D.S. was partially supported by the Israeli Ministry of Science and Technology and by the Edmond J. Safra Center for Bioinformatics at Tel-Aviv University. R.S. was supported by research grants from the Israel Science Foundation (grants no. 715/18 and 757/12). The authors would like to thank Dalya Gartzman and Dr. Roni Wilenchik for their comments on the manuscript.

## Author contributions

R.S. designed the study and provided guidance through the project, D.S. designed the solution, implemented and validated the method. Both authors wrote the manuscript.

## Additional information

**Competing interests:** The authors declare no competing interests.

