## [Peer Review File · Nature Communications]

Reviewers' comments:

Reviewer #1 (Remarks to the Author):

In the presented work, the authors tackle how to assign directionality to protein interactions, an extremely relevant and important problem in systems biology. I acknowledge that this is a challenging problem due to the intrinsic heterogeneity /biases of biological data sets as well as potential context dependency of such directionality. Therefore, I find the authors' supervised learning approach to score the interactions baser on how likely they are to connect cause and effect proteins in the interaction network given a certain data a set is a neat way of tackling the problem. That being said, in its current form, the text is not entirely clear and there are certain issues that need to be addressed.

Major:

1. The results section could use a reorganization of the text in subsections (e.g., Diffuse2Direct: A diffusion based approach to orient interactions, Validation of D2D, etc.). Moreover, the first paragraph could provide a bit more info on the methodology and data sets used to guide reader following the results without having the need to dig into the methods. A few sentences on the data sets and the features (scores coming from network-based diffusion from cause / effect proteins) would help.

2. In my opinion, the so called small scale example is more appropriate as a case study (e.g., at the end of the results). Instead, the authors can focus on a certain interaction between two nodes (e.g., VEGF, VEGFR2 or PI3K) and explain how their method would predict the direction of the interaction using --for instance-- drug response data set. I also find Figure 2 inconsistent with the supplementary network data provided as the directionality of VEGF-VEGFR2 and VEGFR2-Src are reversed in the provided data set and the interactions between VEGFR2 and Cdc42 & PI3K are missing (assuming that the first column is the source node and the second one is the target node in the file). A legend on the meaning of the colors as well as arrow representations should be added to the Figure.

3. I had difficulties understanding the method and its validation due to (i) the use of different names on the scores the authors defined and (ii) the lack of clear definition of the training and test data sets used across different experiments throughout the text:

- Diffuse2Direct(u, v) is not formally defined in the text, I suspect the ratio of $D2D(u, v)$ to $D2D(v, u)$ is the same as $score(u, v)$

- What does "confidence ratio" refer to in Pg10:237?

- It is not clear whether /how the smoothing parameter, α , in $F(v)$ is optimized (and what is the value used in the experiments). Same goes for the value of ϵ that was chosen as 0.1.

- Typically number of folds in cross validation is chosen as 5 or 10, but the authors use 3-fold, does the number of folds have any effect on the accuracy of the presented method?

- What is the training data set / "guiding sources" for the KEGG VEGF pathway?

- What are the training / test data sets used for the results in Table S1?

- "where the balancing parameter is chosen by 3-fold cross validation within the training set".

Again what is the training set used in this case (and what is its value)? Is it the 33,756 interactions with known directions within the "training set"? Part of the confusion arises because AFAIU this data set is also used as a test set in other experiments. ("by using all training data (i.e. using the five test sets...) and integrating all available guiding sources").

- Can the authors use an independent validation set, e.g., directed interactions in KEGG pathways or data set from PATHLINKER (Ritz et al., 2016, Npj Sys Bio Appl) that are not already in their existing test set and show the accuracy of the consensus network on recovering these interactions?

4. The authors mention that they have created the test set by subsampling from the training set but it is not clear whether these two sets are disjoint (not overlapping). Furthermore, how the

sampling of interactions that aim to correct for degree bias is done is not explained in the text. Also, could the authors account for the degree bias when they calculate the diffusion score such that the method would be robust against potential biases in the underlying networks. Indeed, the diffusion formula resembles PageRank with priors algorithm which favors hubs and DADA (Erten et al., 2011, Biodata Min) or NetScore (Guney and Oliva, 2012, Plos One) could be more appropriate to avoid the bias.

5. In Fig 5a 1/3 of the proteins in the interaction network are ranked better than known drug targets, weakening the argument on the potential use of directed interaction network as a reliable tool for drug target prediction. It also creates a dichotomy between existing works that suggest the use of protein interaction networks to predict drug targets (Luo et al., 2017, Nat Comm) or understand drug effect (Guney et al., 2016, Nat Comm) or identify important nodes in the network (Vinayagam et al., 2016, PNAS). It would be interesting to see what would be the result of ranking proteins randomly in the same figure. As much as, I understand that the authors aim to show the improvement compared to undirected/unoriented network, it would be useful to demonstrate an application supporting the practical utility of the directed network.

6. Similarly, on the prediction of driver genes, the authors might consider showing advantage of using directed interaction network in combination with HotNet2 (Leiserson et al., 2014, Nat Genet) or Paradigm-SHIFT (Ng et al., 2012, Bioinformatics), though I understand that this could be out of the score of the current work. On the other hand, given that driver genes across various cancer types are shared, the authors might want to check whether the other cancer data sets and positive controls do not contain the driver genes in the left-out AML data set. It would also be interesting to see the same analysis when the other cancer data sets are left out individually.

7. Finally, I find the method presented by the authors and the data sets used to validate it of very high value to the community and to maximize its benefit and allow its reproducibility, I encourage them to make the code and data used in their analyses publicly available. Given the context specific nature of the directionality of interactions, this would greatly help researchers to apply the methodology on their own data sets.

Minor:

Several typos / exerts that are unclear

- "diverse disease settings"
- Pg3:67 "These proximities ... these scores"
- Pg3:73 "directed part / undirected part"
- Pg4:82 "the effect .. as emanating from .. and affecting the genes..."
- FAX-Paxilim  FAK
- Pg9:218 Ref 34 should be 37
- Pg11:264 "we tried a variant of the method" what was different in this method?
- "guiding source" can be explicitly defined at its first appearance to avoid confusion
- Ref 38 lacks journal info
- "paired sets of causes and effects" / "paired effect proteins"

Reviewer: Emre Guney

Reviewer #2 (Remarks to the Author):

Silverbush and Sharan present an approach called Diffuse2Direct for orienting edges in a PPI network. Diffuse2Direct takes the following as input: an undirected or mixed (containing both undirected and directed edges) weighted PPI network and a collection of experiments (here, drug response data from CMap), each of which induces a set of causal proteins (drug targets) and a set of effect proteins (differentially expressed genes). For each experiment in the collection, Diffuse2Direct computes a diffusion score for each edge based on two random walks with restarts,

one from causal proteins and another from effect proteins on the same network, resulting in a feature vector for every edge (columns are scores obtained for that edge for different experiments). These features are then used to train a logistic regression classifier to obtain a confidence score for every edge direction. The authors then show that their method outperforms a method developed by Vinayagam et al. (I will refer to it as "SPC"), the SHORTEST algorithm proposed by the authors of this manuscript, and a method called DADA.

The method itself is new but not surprising. Orienting the human protein interaction network is valuable to the network biology community. The authors have several results and performed multiple comparisons, some of which showcase the power of the new method. Nevertheless, crucial aspects of the analysis have been left out or are confusing. I have the following concerns and several suggestions for additional analysis or clarifications.

A high level concern with the manuscript is that the presentation of the motivation for the proposed method could be improved. I buy the idea of two random walks and edge scores based on them (subject to some comments below). But after that "outsourcing" the final result to a logistic regression based classifier seems to be poorly justified. It raises the question of how much of the quality of the final results comes from the random walks and how much from the logistic regression. I hope that points 4 and 6 below will help to address this important matter.

1. This concern is minor but I have put it first since it relates to the name of the approach. Isn't the "Diffuse" algorithm just random walk with restarts to a subset of nodes? It will make it clearer to label the algorithm "RWR2Direct" and give the nod explicitly to the well-known and widely used RWR algorithm. Moreover, they can then refer to $1-\alpha$ as the restart probability to any node in the prior set.

2. I am unclear about the rationale for the approach. It seems to me that if the edge is directed from u to v , then the ratio $F^E(v)/F^E(u)$ will be greater than 1 if F^E is computed after reversing every edge in the interactome (a la the TieDie method, <https://www.ncbi.nlm.nih.gov/pmc/articles/PMC3799471/>). Since the interactome contains some directed edges, I do not see why $F^E(v)/F^E(u)$ will be greater than 1 as calculated.

3. If the authors indeed meant what they said, they can easily compare their approach to this modified version, where they perform the second RWR after reversing the edges in the graph (after changing the definition of the score appropriately). These scores can be fed into the logistic regression classifier. I suspect that the results will not change much since the number of directed edges is much smaller than the number of undirected graphs, but it will be useful to add these results for the sake of completeness.

4. The authors should also include a comparison to the eQED algorithm (<http://msb.embopress.org/content/4/1/162.long>). I would like to see using a score based on the voltage differences across the edges fed into the logistic regression classifier. This approach will test whether the power of the new approach comes from the RWR or from the logistic regression.

5. The results from the DADA algorithm are only in Table S1 in the supplement. Including the precision recall curves in Figure 3 will be informative to the reader. It will also help to refer to this algorithm as DADA in line 94.

6. Figure S2 is a nice result but is also a black box result, since we do not get a sense of how the classifier is selecting features. Can the authors peer inside the classifier and see some trends on which features are heavily weighted? They use L1 regularization so I would expect many features to have scores close to 0. Can the authors get some information about features that are used consistently or features that actually contribute to the results as the number of features increases? For example, if a drug diffuses into the cell and binds directly to a nuclear receptors, I expect it will not help much with direction prediction. I know the analysis is challenging and what I have

requested is not concrete but I hope the authors will appreciate that this suggestion is made with the intention of helping to improve the manuscript.

7. The analysis to the KEGG VEGF pathway is nice in terms of illustrating how the method works. However, there are several points that need clarification here.

a. The authors should clarify that they use only the results from the two RWRs here (lines 75–80), i.e., they do not use the logistic regression classifier. This clarification is important since the later plots and figures in the results use the score from logistic regression. Adding to the confusion is a final interactome confidence that ranges from 1 to 5.

b. They should state the sources and targets for the VEGF analysis.

c. A larger issue here that has to do with the way the KEGG databases combines information in a node. Each node can correspond to a set of proteins, e.g., a protein family. For example, PI3K (bottom left in Figure 2) corresponds to six unique NCBI Gene ids. How did the authors deal with mapping this node and the corresponding six genes to the node in their interactome? I have found parsing KEGG pathways to be difficult even with the greatest of care. Without additional information on the decisions made by the author, it is not possible to ascertain the value of this result.

d. What did the authors do to be able to include phrases such as “Focal adhesion turnover” and “Actin reorganization” in their network?

e. It is unclear how the Calcium ion can be a node in their network (“Ca” in Figure 2).

f. Note that the protein towards the top left is “VRAP” and not “VARP”.

8. I liked the results in Figure 3 since they show the improvement over Vinayagam for a number of diverse datasets. There are other valuable sources of pathway oriented directed interactions that the authors should measure the performance of their algorithms. KEGG and NetPath are two such leading databases. I suggest avoiding KEGG due to the difficulties caused by protein-family-based nodes and the idiosyncrasies of the way KEGG records interactions. Therefore, the authors should attempt to predict the interactions in each NetPath pathway, given the sources (receptors) and targets (transcription factors) in each pathway. They can aggregate the results across all the pathways.

9. I have the following additional concerns with the comparison shown in Figure 3 to the SPC method proposed by Vinayagam et al.

a. SPC uses a seven-feature vector for each edge that is independent of experimental data as acknowledged by the authors in lines 260-264. The authors state that “In addition, we tried a variant of the method guided by drug response data, yet this led to a low recall.” However, they do not provide any details of this modification or show their results after integrating drug response data with SPC, making it difficult to judge the improvement obtained by their approach. It is unclear why the statement “yet this led to a low recall” should hold true for SPC. According to Figure 2A in the paper by Vinayagam et al., they make a prediction for each edge direction, so for some cut-off value, the recall should be 1 irrespective of the dataset used.

b. The analysis shown in Figure 3 is said to be for 3-fold CV with 50% correct orientations and 50% incorrect orientations (according to Table S1). It is unclear how the authors obtained 50% incorrect orientations in each case. In other words, what were the sources of edges for the positives and negatives in each fold? For example, there are 117 directed edges in EGFR pathway. In three fold cross validation, I assume that 39 edges belong to the positive set held out in each fold. What about negative edges? Are the incorrect orientations meant to provide negative sets?

c. Further, the choice of 3 folds seems arbitrary. A natural idea is to try other folds, say 2, 4, and 5. It would be enough to perform this analysis only for the algorithm proposed by the authors.

10. The authors have taken the trouble to show that the oriented network can be used to find functional roles of proteins. Results appear in Figure 5. I do not find these results to be particularly exciting. In Figure 5a, the improvement in the mean rank is quite modest, even if statistically significant. I would be hard-pressed to say that the proposed method is predicting valuable information. I urge the authors not to use the word "prediction" to describe these types of results.

11. I find the improvements of the oriented network over the unoriented network in Figure 5b to also be quite modest. The text in lines 164–173 is somewhat misleading. For example, when the authors say "the orientation based computation reports 49% of the AGO positive list consisting of 1429 genes" what they really mean is that "49% of the top 1% of the genes reported by the orientation-based computation are in the AGO positive list". The denominator is the number of genes in the top 1% of results and not the genes in the positive list. The authors should rephrase all the sentences in this part accordingly.

12. In addition, I am not convinced by this analysis. What prompted the authors to select just the top 1% of genes? The positive lists contain at least 500 and even 1000s of genes. I can imagine a different analysis where the authors plot the fraction of the overlap between the positive set and the top k% of genes, for k varying between, say 1 and 50. Or they could compute a full precision-recall curve. There are many options here. I see the value these post-hoc analyses add to the paper, but as described I do not find them very useful.

13. The approach has at least two parameters: alpha, where $1 - \alpha$ is the RWR restart probability, and epsilon. I think they refer to alpha as the balancing parameter in line 238. It will help to name the parameter explicitly in that line. It will be very useful to give details on the regularization approach and how it fits with the logistic regression framework. What values of alpha did the regularization yield?

14. Could the authors not have use the cross validation framework to set the value of epsilon as well rather than fix it arbitrarily at 0.1?

Other Concerns:

1. Please use mathematical notation for the p-values, e.g., in line 151, the p-value as written seems to be "e to the power -12" rather than 10^{-12} .

2. The notation to describe the diffusion process can be improved. In line 216, the authors have not defined the set Priors. It will be so much cleaner if they first defined this set and then explained the diffusion process. The equations will look nicer if the authors use symbols instead of words, e.g., $w(u,v)$ instead of $\text{weight}(u,v)$. The weight term with a caret above it looks inelegant. Similarly, use P for the set of prior proteins. It will also help to explain in what sense these proteins are priors.

3. In lines 217-218, citation 34 is a paper by Yosef et al. Moreover, is there a need to cite a paper here unless the authors are using something other than standard power iteration?

4. In lines 220-226, what does "computing a propagation" mean? That phrase is unclear. Perhaps the authors mean to say "We apply the diffusion procedure once by setting the causal proteins to be the set P of prior proteins. We use $F^C(u)$ to denote the score computed for each node u ." Later, they can say "We apply the diffusion procedure a second time after setting the set P to contain the effect proteins. We use $F^E(u)$ to denote the score computed for each node u ."

5. For the drug response data, did CMAP provide the drug target information as well?
6. In line 442, the spelling of "parentheses" is wrong.
7. The 3D bars in Figure 4b and in Figure S3 seem unnecessary. 2D bars should be sufficient.
8. The authors say that the "unoriented" network is the one with all edge directions "flipped" compared to oriented network (lines 146-148, 154-156), which makes me think that the edge directions are simply reversed. However, the icon below the first bar in Figure 5a displays what appears to be mostly an undirected network as "unoriented network." Is the icon incorrect? The icons also include some colored shapes that the authors do not explain. The authors should consider simply deleting this icon.
9. It is unclear how the authors compute "orientation_confidence" column in Supplementary Information file. Are they the number of "votes for each direction from the five source-specific directed networks" (lines 139-140)? However, some edges have scores of 1.5. Moreover, won't the number of votes change based on the value of epsilon? I assume they used 0.1 here.

Reviewer #3 (Remarks to the Author):

The idea of orienting PPI networks by diffusing signals from causal proteins to affected proteins is good. The results seem interesting: I particularly liked figure 2 and Figure 4.

I feel that the authors could perhaps make clearer what the main message of this paper is. The main finding of this work appears to be that an oriented PPI network is better than an unoriented one at predicting drug targets and cancer driver genes. This is, in fact, one of the results of the paper – in lines 142-175 the authors show that a very simple ranking method (based on proximity) gives better results at predicting drug targets and cancer driver genes when using an oriented PPI (oriented using the proposed method) than an un-oriented one. However, the title and the abstract could be interpreted as the authors proposing a method for improving the state-of-the-art prediction of drug targets and cancer driver genes – this is not the case here, as in order to show that, the authors would need to compare performances against state-of-the-art methods for these problems. At the same time, most of the paper is about showing that the proposed method for orienting edges provides a better orientation than existing methods. Should this be the main message of this paper?

Many details are missing; I don't think we would be able to implement the algorithms described here. It would be useful to provide the datasets and the code which was run in the experiments.

More specific comments:

(1) When reading the introduction, it feels like the authors are suggesting that the entire PPI network should be oriented. While the authors do refer to signalling circuitry, I found sentences like the one on lines 33-35 a bit misleading ("Nevertheless, to date, direction information is available for only a small percentage of the interactions. For example, KEGG contains 5,769 directed interactions in human out of the current 311,962 interactions present in BioGRID"). This issue becomes then clear later (lines 72-74), but I believe it would be easier to have a clearer overview in the introduction (and possibly in the abstract).

(2) The result on page 4, figure 2, is very good. But I have a couple of points:

a) There is only one short paragraph describing the experiment, at the beginning of page 4 with almost no details. Could the authors provide more details? Which datasets were used? How were parameters tuned? Etc.

b) More importantly, it would be interesting to understand whether such a good recovery of directionality for this pathways is an isolated case or whether it works in general. Would it be possible to quantify the method's performance on the entire KEGG? This would be much more convincing.

(3) About the second results, "drug response to orient a human PPI network", starting on line 81.

a) In Table S1, the authors should expand what they mean by "Instances (50% correct orientations + 50% incorrect orientations)" – it is not easy to follow, and it becomes clear only when one connects it to the fact that there are 2 entries for each connection in the output of their method.

b) In the same S1 table, I believe that the last column ("Benchmark orientation") refers to the DADA approach, but it is not clear.

c) Figure 3: why some of the green curves don't reach 0.5?

(4) About the third result, "using gene pairs derived from genomic mutation data and the resulting expression changes" to orient a human PPI network, starting on line 120: it would be useful to compare the result with the methods of Vinayagam and SHORTEST, showing AUPR curves – just as it was done for the previous (second) set of results.

(5) About the fourth result, starting on line 142.

I could not understand what the authors mean here: "To this end, we flipped all directions in the network and computed a network diffusion score using the differentially expressed genes induced by the drug as a prior set."

In the same way, on line 154, this sentence was not clear: "As before we flipped the network, and computed the network diffusion scores for each patient separately, diffusing from the patient differentially expressed genes."

My understanding is that the authors are testing whether an oriented network is better than an unoriented one at predicting drug targets and cancer driver genes, but it would be easier for the reader if they expanded their explanation.

(6) In general, the output of the proposed method is a partially oriented graph, where some edges are directed, and some undirected. The authors tested and quantified the correctness of the directed edges. I feel that it would make sense to also test and quantify the correctness of the edges that are left undirected by the algorithm. Test sets for this could be obtained, for example, from known protein complexes.

(7) It would be important to have a more detailed explanation of the diffusion method – explained beginning in line 210. As it stands, I don't feel the paper is self-contained – it refers to explanations from the Cowen et al paper which I think would be good to include here.

Also, in the explanation on page 9, it was unclear to me: how many iterations is the algorithm run for? What is the "prior set"?

It would also be good to include some intuition for the method.

(8) In general, 3-fold cross validation is not ideal – it is not too far from just using training and testing, and no validation. Why did the authors not use the more standard 10-fold cross validation? I suspect that the numerosity of the dataset did not allow for it, but I feel it would be useful to discuss this point.

(9) Lines 235-240 were not clear to me. Many details are missing. (Again, the formulas for the logistic regression classifier with L1 regularization (possibly in the supp mat) would make the paper more self-contained). Which parameters are learned through a 3-fold cross validation? Does

the “balancing parameter” on line 238 refer to the alpha parameter on line 214, or does it have to do with the L1 regularization? Where does value of 0.1 for epsilon come from?

(10) The sentence “we filtered the test sets by random down-sampling so that similar numbers of interactions from high to low degree and from low to high degree proteins remain.” was not very clear to me. Could you expand? Could you describe the algorithm in detail? (it could go in the supplementary material)

MINOR COMMENTS

Fig 1a, green and blue circle could be used in the different layers

Line 218, the reference should probably be 37

Reviewer 1

Major:

1. The results section could use a reorganization of the text in subsections (e.g., Diffuse2Direct: A diffusion based approach to orient interactions, Validation of D2D, etc.). Moreover, the first paragraph could provide a bit more info on the methodology and data sets used to guide reader following the results without having the need to dig into the methods. A few sentences on the data sets and the features (scores coming from network-based diffusion from cause / effect proteins) would help.

We have revised the Results section as requested.

2. In my opinion, the so called small scale example is more appropriate as a case study (e.g., at the end of the results). Instead, the authors can focus on a certain interaction between two nodes (e.g., VEGF, VEGFR2 or PI3K) and explain how their method would predict the direction of the interaction using --for instance-- drug response data set.

The small scale example serves to exemplify the propagation part of the method and is disjoint from all subsequent analyses and datasets used, hence we believe it fits best in the first subsection of the results and is more convincing than using a single interaction. We have now extended both the text and the corresponding Figure 2 to more clearly explain and exemplify our method using multiple curated pathways.

I also find Figure 2 inconsistent with the supplementary network data provided as the directionality of VEGF-VEGFR2 and VEGFR2-Src are reversed in the provided data set and the interactions between VEGFR2 and Cdc42 & PI3K are missing (assuming that the first column is the source node and the second one is the target node in the file).

We now clarify in the text that the small scale network was taken from KEGG and its orientation was based on the KEGG pathway structure. Our physical network was taken from other sources and its orientation was based on large scale genomic data. Hence, interactions could potentially be missing/different. Regarding the specific interactions mentioned, we couldn't find VEGFR2 in the consensus network – which Entrez ID was the reviewer using for it? Regardless, we improved the annotation of the supplementary network data changing the annotation to source entrez id and target entrez id.

A legend on the meaning of the colors as well as arrow representations should be added to the Figure.

Fixed.

3. I had difficulties understanding the method and its validation due to (i) the use of different names on the scores the authors defined and (ii) the lack of clear definition of the training and test data sets used across different experiments throughout the text:
- Diffuse2Direct(u, v) is not formally defined in the text, I suspect the ratio of D2D(u, v) to D2D(v, u) is the same as score(u, v)

We apologize for the lack of a clear name for our method's scores. score(u,v) denotes the score assigned to a given direction based on a specific guiding source (feature) via diffusion. The final score of a direction combines the information from all features using a classifier. We have now named this final score D2D(u,v) (previously it was Diffuse2Direct(u,v)).

- What does “confidence ratio” refer to in Pg10:237?

We have now explicitly written the formula for this ratio of the D2D score in one direction and the D2D score in the other direction (p. 13):

To direct an edge, we choose the highest scoring direction unless the ratio of both scores, $\text{Max}\{\frac{D2D(u,v)}{D2D(v,u)}, \frac{D2D(v,u)}{D2D(u,v)}\}$, is below $1 + \epsilon$, where we set ϵ to be 0.1.

- It is not clear whether /how the smoothing parameter, alpha, in F(v) is optimized (and what is the value used in the experiments). Same goes for the value of epsilon that was chosen as 0.1.

We thank the reviewer for this comment. We chose the value for the smoothing parameter based on previous publications (Leiserson et. al. Nature Genetics 2015), showing that network diffusion is generally robust to any value within the range of 0.4 to 0.6. We tested the robustness of our findings to the value of the smoothing parameter and found that the precision-recall results were robust to parameter values in the range 0.1 to 0.9, with a slight preference to intermediate values (0.3 or 0.6 which is the value used in the paper). Such values balance between the prior knowledge and the network structure. We added the analysis to the Supplementary Information, Figure S1b.

Regarding the value of epsilon, which controls the number of interactions that are oriented, we chose it so that approximately 1/3 of the interactions remain unoriented in the consensus network, as suggested in ref. 14. Following the reviewer's comment we show that the observed enrichment of unoriented interactions with intra-complex interactions is robust to the value of epsilon used in a wide range from 0.05 to 0.15. We added this information to the Methods and Figure S1c.

- Typically number of folds in cross validation is chosen as 5 or 10, but the authors use 3-fold, does the number of folds have any effect on the accuracy of the presented method?

We chose 3-fold for efficiency. As requested, we repeated the analyses with 5 and 10 folds and found that the results are consistent with a slight increase in performance as the number of folds increase. We added the analysis to the Supplementary Information, Figure S1a.

- What is the training data set / “guiding sources” for the KEGG VEGF pathway?

For the small scale example, the guiding sources were based on the pathway's structure, with causes being proteins with no in-going edges (input nodes) and effects being proteins with no out-going edges (output nodes). We now clarify this in the pertaining paragraph in the Results (p. 4).

- What are the training / test data sets used for the results in Table S1?

We thank the reviewer and added an explanation (p. 5) that each interaction type was tested separately in 3-fold cross validation.

- “where the balancing parameter is chosen by 3-fold cross validation within the training set”. Again what is the training set used in this case (and what is its value)? Is it the 33,756 interactions with known directions within the “training set”? Part of the confusion arises because AFAIU this data set is also used as a test set in other experiments. (“by using all training data (i.e. using the five test sets...) and integrating all available guiding sources”).

We thank the reviewer for pointing out the confusion and revised the Inference of directions Section of the Methods to include the following information (p. 13): To avoid over fitting and restrict the number of features, we used L1 regularization where the value of the regularization balancing parameter was chosen via a nested 3-fold cross-validation (i.e., the parameter value is decided by cross validation within the training set, and then applied to the test set) in the range of 10^{-4} to 10^4 .

- Can the authors use an independent validation set, e.g., directed interactions in KEGG pathways or data set from PATHLINKER (Ritz et al., 2016, Npj Sys Bio Appl) that are not already in their existing test set and show the accuracy of the consensus network on recovering these interactions?

We thank the reviewer for this wonderful suggestion. We compared the consensus network to PathLinker (which includes also directions from KEGG) and found an overall high agreement: Out of 33% (69,704) of the network's edges that were oriented with maximum confidence, 1917 were included in Pathlinker with 70% agreement (1334 had the same orientation and 583 were oriented differently, hypergeometric p -value $<1.33e-91$). Out of 69% of the edges that were oriented with confidence at least 2, 2815 were included in PathLinker with 63% agreement ($p<2e-67$). We added these results to the main text (p. 8).

4. The authors mention that they have created the test set by subsampling from the training set but it is not clear whether these two sets are disjoint (not overlapping).

All test sets were disjoint from the corresponding training sets. The downsampling refers to eliminating the degree bias prior to selecting the training and test portions of a given set.

Furthermore, how the sampling of interactions that aim to correct for degree bias is done is not explained in the text.

We added the required information to the Methods (p. 16).

Also, could the authors account for the degree bias when they calculate the diffusion score such that the method would be robust against potential biases in the underlying networks. Indeed, the diffusion formula resembles PageRank with priors algorithm which favors hubs and DADA (Erten et al., 2011, Biodata Min) or NetScore (Guney and Oliva, 2012, Plos One) could be more appropriate to avoid the bias.

Indeed, degree bias is an important concern when orienting a PPI network. The scores we use to orient the network do not suffer from this bias, as each node's score appears twice – once in the numerator and once in the denominator:

$$score(u, v) = \frac{F^C(u) \cdot F^E(v)}{F^C(v) \cdot F^E(u)}$$

thus canceling any potential bias. We also tested the performance of orienting the network according to its topology only using network topology. This explicitly benchmarks the bias of our method and the test sets we use. We found that using the network topology only yields substantially lower performance (Table S1), supporting our scoring scheme. We added this discussion to the Methods (p. 12).

5. In Fig 5a 1/3 of the proteins in the interaction network are ranked better than known drug targets, weakening the argument on the potential use of directed interaction network as a reliable tool for drug target prediction. It also creates a dichotomy between existing works that suggest the use of protein interaction networks to predict drug targets (Luo et al., 2017, Nat Comm) or understand drug effect (Guney et al., 2016, Nat Comm) or identify important nodes in the network (Vinayagam et al., 2016, PNAS). It would be interesting to see what would be the result of ranking proteins randomly in the same figure.

Random ranking produces uniform ranks – we added this ranking to Figure 5 as requested.

As much as I understand that the authors aim to show the improvement compared to undirected/unoriented network, it would be useful to demonstrate an application supporting the practical utility of the directed network.

Our intention was to give a proof of concept example for the utility of a directed network; indeed we gave two such examples, of ranking drug targets and ranking cancer driver genes. Going deeper into these/other examples would be out of scope.

6. Similarly, on the prediction of driver genes, the authors might consider showing advantage of using directed interaction network in combination with HotNet2 (Leiserson et al., 2014, Nat Genet) or Paradigm-SHIFT (Ng et al., 2012, Bioinformatics), though I understand that this could be out of the score of the current work.

Please see the previous answer.

On the other hand, given that driver genes across various cancer types are shared, the authors might want to check whether the other cancer data sets and positive controls do not contain the driver genes in the left-out AML data set.

We note that the left-out disease (AML in this example) driver genes were not input to the algorithm, rather predicted by it based on the learned orientation. Following the reviewer's concern we tested whether any of the cancer studies used in the paper is biased towards the control lists. We therefore compared the mutations frequency distribution of each cancer to each control list. Reassuringly, we found that genes that are driver genes (i.e. in one of the control lists) have similar mutation frequency distribution as genes that are not in the list. Two-sample Kolmogorov-Smirnov tests between each cancer and each control list yielded insignificant p-value, ranging from 0.9 to 0.98, with the lowest obtained for Breast cancer and the "Text mining" driver list.

It would also be interesting to see the same analysis when the other cancer data sets are left out individually.

We thank the reviewer for this suggestion, indeed we found that throughout the different cancer types the directed network consistently recovers more known driver genes than the undirected network (with similar false positive rates for both types of networks). Our findings are summarized in the Supplementary information (p. 8, Figure S5).

7. Finally, I find the method presented by the authors and the data sets used to validate it of very high value to the community and to maximize its benefit and allow its reproducibility, I encourage them to make the code and data used in their analyses publicly available. Given the

context specific nature of the directionality of interactions, this would greatly help researchers to apply the methodology on their own data sets.

We now published the code as an open source: <https://github.com/danasilv/Diffuse2Direct>

Minor:

Several typos / exerts that are unclear

- “diverse disease settings”
- Pg3:67 “These proximities ... these scores”
- Pg3:73 “directed part / undirected part”
- Pg4:82 “the effect .. as emanating from .. and affecting the genes...”
- FAX-Paxilim  FAK
- Pg9:218 Ref 34 should be 37
- Pg11:264 “we tried a variant of the method” what was different in this method?
- “guiding source” can be explicitly defined at its first appearance to avoid confusion
- Ref 38 lacks journal info
- “paired sets of causes and effects” / “paired effect proteins”

All minor comments were addressed.

Reviewer 2

A high level concern with the manuscript is that the presentation of the motivation for the proposed method could be improved. I buy the idea of two random walks and edge scores based on them (subject to some comments below). But after that “outsourcing” the final result to a logistic regression based classifier seems to be poorly justified. It raises the question of how much of the quality of the final results comes from the random walks and how much from the logistic regression. I hope that points 4 and 6 below will help to address this important matter.

We extended the small scale example to better demonstrate that the diffusion is powerful on its own when a single source of information is available (p. 4-5 and Figure 2). We now explicitly explain that we use a classifier on top of the diffusion in order to consolidate diffusion scores that are based on different information sources (causal and affected protein sets).

1. This concern is minor but I have put it first since it relates to the name of the approach. Isn't the “Diffuse” algorithm just random walk with restarts to a subset of nodes? It will make it clearer to label the algorithm “RWR2Direct” and give the nod explicitly to the well-known and widely used RWR algorithm. Moreover, they can then refer to $1-\alpha$ as the restart probability to any node in the prior set.

As the reviewer points out, the diffusion/random walk with restart method comes in different flavors and different names. We have recently written a review highlighting the commonalities between these different variants [ref. 41 in the current manuscript]. Here we chose to name our method diffusion to make the name more accessible and less technical.

2. I am unclear about the rationale for the approach. It seems to me that if the edge is directed from u to v , then the ratio $F^E(v)/F^E(u)$ will be greater than 1 if F^E is computed after reversing every edge in the interactome (a la the TieDie method, <https://www.ncbi.nlm.nih.gov/pmc/articles/PMC3799471/>). Since the interactome contains some directed edges, I do not see why $F^E(v)/F^E(u)$ will be greater than 1 as calculated.

Indeed, as the reviewer points out, edge directions are reversed in the diffusion from the effects. We clarified the text accordingly and plotted the distribution of scores for an example pathway in Figure 2c. See also our next answer.

3. If the authors indeed meant what they said, they can easily compare their approach to this modified version, where they perform the second RWR after reversing the edges in the graph (after changing the definition of the score appropriately). These scores can be fed into the logistic regression classifier. I suspect that the results will not change much since the number of directed edges is much smaller than the number of undirected graphs, but it will be useful to add these results for the sake of completeness.

Prior knowledge about an edge direction ($u \rightarrow v$) can be used either in the diffusion process – by keeping the directions of the training edges fixed while diffusing, or in the classifier – by using the D2D score vector for training, but not for both. To test the reviewer's suggestion, we partitioned the training edges to those that are used for the diffusion and all others. Specifically, we used the known orientation for four datasets during the diffusion (KPIs, PDIs, EGFR, E3) and left the fifth, STKE, unoriented. We then tested the performance of our method in 3-fold cross validation on this last test set. As the reviewer suspected, we found that the results did not

change. We also tried a different version where 2/3 of the STKE edges remained directed during the diffusion, and only 1/3 of the edges were subjected to a 3-fold cross validation. This version also yielded similar (slightly worse) results. As these versions require extra diffusion computations (since different parts of the network are kept directed every time) we maintained our original version of the method and explained our choice in the text (including the effect on the results in the Supplement, p. 1).

4. The authors should also include a comparison to the eQED algorithm (<http://msb.embopress.org/content/4/1/162.long>). I would like to see using a score based on the voltage differences across the edges fed into the logistic regression classifier. This approach will test whether the power of the new approach comes from the RWR or from the logistic regression.

eQED is a method for gene prioritization. While it is possible to use its output also for direction prediction, the method was applied only in yeast and its code is not readily available, hence we could not directly compare to it. We note that eQED is again a variant of diffusion [as stated in the paper itself and also reviewed in ref. 41], hence we expect it to yield similar raw scores (per guiding source used).

5. The results from the DADA algorithm are only in Table S1 in the supplement. Including the precision recall curves in Figure 3 will be informative to the reader. It will also help to refer to this algorithm as DADA in line 94.

We included the benchmark results in Figure 3 as suggested by the reviewer. Since the method is based on a correction presented in the DADA paper, but is not exactly DADA, we refer to it as a topology-based benchmark.

6. Figure S2 is a nice result but is also a black box result, since we do not get a sense of how the classifier is selecting features. Can the authors peer inside the classifier and see some trends on which features are heavily weighted?

We thank the reviewer for this important suggestion! We quantified the number of times a drug had been used as a contributing feature, separately for each test set and when aggregating the overall use of the feature as suggested in the following comment. We calculated the correlation of the use of the drug as a feature with 1) number of known drug targets, 2) number of genes that were observed to be differentially expressed in response to the drug, 3) cellular location of known drug targets, and 4) cellular location of the genes that were observed to be differentially expressed in response to the drug. We discuss the results in the main text pages 6-7 and Figure S4.

They use L1 regularization so I would expect many features to have scores close to 0. Can the authors get some information about features that are used consistently or features that actually contribute to the results as the number of features increases? For example, if a drug diffuses into the cell and binds directly to a nuclear receptors, I expect it will not help much with direction prediction. I know the analysis is challenging and what I have requested is not concrete but I hope the authors will appreciate that this suggestion is made with the intention of helping to improve the manuscript.

Indeed, as expected by the reviewer, we found that the cellular localization of the drug targets is the most significant characteristic of whether the drug contributes to the orientation. In particular, drugs with targets in the membrane were the most informative for the orientation. The

next informative characteristic was the number of known drug targets and location in the extracellular matrix. Interestingly, although not statistically significant, the fourth informative characteristic was the location of the observed effect of the drug, where drugs that had a substantial number of differentially expressed genes located in the nucleus were more informative. We added these results to the text (p. 7).

7. The analysis to the KEGG VEGF pathway is nice in terms of illustrating how the method works. However, there are several points that need clarification here.

a. The authors should clarify that they use only the results from the two RWRs here (lines 75–80), i.e., they do not use the logistic regression classifier. This clarification is important since the later plots and figures in the results use the score from logistic regression. Adding to the confusion is a final interactome confidence that ranges from 1 to 5.

Indeed the reviewer is right and we clarified the text accordingly as well as how we compute the confidence values of the consensus network (p. 8).

b. They should state the sources and targets for the VEGF analysis.

Thanks. We have clarified that the sources are nodes with no in-going connections in the pathway and the targets are nodes with no out-going connections (p. 4).

c. A larger issue here that has to do with the way the KEGG databases combines information in a node. Each node can correspond to a set of proteins, e.g., a protein family. For example, PI3K (bottom left in Figure 2) corresponds to six unique NCBI Gene ids. How did the authors deal with mapping this node and the corresponding six genes to the node in their interactome? I have found parsing KEGG pathways to be difficult even with the greatest of care. Without additional information on the decisions made by the author, it is not possible to ascertain the value of this result.

For the small scale example we used a network that consists of the KEGG pathway only, that is, the network's nodes were the pathway's entities as is, which could represent proteins, other molecules and even cellular processes. We now clarify this in the text (p. 4),

d. What did the authors do to be able to include phrases such as “Focal adhesion turnover” and “Actin reorganization” in their network?

Please see the previous answer.

e. It is unclear how the Calcium ion can be a node in their network (“Ca” in Figure 2).

Please see the previous answer to item c.

f. Note that the protein towards the top left is “VRAP” and not “VARP”.

Fixed.

8. I liked the results in Figure 3 since they show the improvement over Vinayagam for a number of diverse datasets. There are other valuable sources of pathway oriented directed interactions that the authors should measure the performance of their algorithms. KEGG and NetPath are two such leading databases. I suggest avoiding KEGG due to the difficulties caused by protein-

family-based nodes and the idiosyncrasies of the way KEGG records interactions. Therefore, the authors should attempt to predict the interactions in each NetPath pathway, given the sources (receptors) and targets (transcription factors) in each pathway. They can aggregate the results across all the pathways.

We followed the reviewer suggestion of analyzing NetPath and also expanded our evaluation with respect to KEGG. These results are now discussed on pages 4-5 and depicted in Figure 2b.

9. I have the following additional concerns with the comparison shown in Figure 3 to the SPC method proposed by Vinayagam et al.

a. SPC uses a seven-feature vector for each edge that is independent of experimental data as acknowledged by the authors in lines 260-264. The authors state that "In addition, we tried a variant of the method guided by drug response data, yet this led to a low recall." However, they do not provide any details of this modification or show their results after integrating drug response data with SPC, making it difficult to judge the improvement obtained by their approach. It is unclear why the statement "yet this led to a low recall" should hold true for SPC. According to Figure 2A in the paper by Vinayagam et al., they make a prediction for each edge direction, so for some cut-off value, the recall should be 1 irrespective of the dataset used.

We thank the reviewer for pointing out this lack of details. We added the required information to the main text (p. 14): In addition, we tried a variant of the method [Vinayagam et. al.] guided by drug response data, rather than membrane-to-transcription-factor pairs, yet only 16% of the interactions in the network were covered by shortest paths from drug response data, information which the method relies on.

b. The analysis shown in Figure 3 is said to be for 3-fold CV with 50% correct orientations and 50% incorrect orientations (according to Table S1). It is unclear how the authors obtained 50% incorrect orientations in each case. In other words, what were the sources of edges for the positives and negatives in each fold? For example, there are 117 directed edges in EGFR pathway. In three fold cross validation, I assume that 39 edges belong to the positive set held out in each fold. What about negative edges? Are the incorrect orientations meant to provide negative sets?

Indeed, the false orientations (negatives) are simply the opposites of the true orientations (positives). We clarified the text accordingly (p. 16, and SI p. 3 after Table S1).

c. Further, the choice of 3 folds seems arbitrary. A natural idea is to try other folds, say 2, 4, and 5. It would be enough to perform this analysis only for the algorithm proposed by the authors.

We chose 3-fold for efficiency. We now repeated the analyses with 5 and 10 folds and found that the results are consistent with a slight increase in performance as the number of folds increase. We added the analysis to the Supplementary Information, Figure S1a.

10. The authors have taken the trouble to show that the oriented network can be used to find functional roles of proteins. Results appear in Figure 5. I do not find these results to be particularly exciting. In Figure 5a, the improvement in the mean rank is quite modest, even if statistically significant. I would be hard-pressed to say that the proposed method is predicting valuable information. I urge the authors not to use the word "prediction" to describe these types of results.

Indeed these results are meant to serve as a proof-of-concept for the utility of the method. We reworded the text accordingly to avoid the use of the word "prediction".

11. I find the improvements of the oriented network over the unoriented network in Figure 5b to also be quite modest. The text in lines 164–173 is somewhat misleading. For example, when the authors say “the orientation based computation reports 49% of the AGO positive list consisting of 1429 genes” what they really mean is that “49% of the top 1% of the genes reported by the orientation-based computation are in the AGO positive list”. The denominator is the number of genes in the top 1% of results and not the genes in the positive list. The authors should rephrase all the sentences in this part accordingly.

We thank the reviewer for pointing out the needed clarification, and we revised the text as suggested. We also greatly extended the results to include a range of K values for the top K% proteins being examined (Figure S5, see next answer).

12. In addition, I am not convinced by this analysis. What prompted the authors to select just the top 1% of genes? The positive lists contain at least 500 and even 1000s of genes. I can imagine a different analysis where the authors plot the fraction of the overlap between the positive set and the top k% of genes, for k varying between, say 1 and 50. Or they could compute a full precision-recall curve. There are many options here. I see the value these post-hoc analyses add to the paper, but as described I do not find them very useful.

We thank the reviewer for this great suggestion. We now added tests of a range of possible Ks (Figure S5) for all four disease sets (AML, breast, colon and ovarian cancer). We found that the oriented network consistently reports more driver genes and less non-driver genes compared to the unoriented one. We used a range of 0.25% to 20%, as after 20% the differences flatten. We also include at the end of the letter the precision-recall curves, although we chose to omit them from the manuscript in favor of the current summary figure (Figure S5).

13. The approach has at least two parameters: alpha, where $1 - \alpha$ is the RWR restart probability, and epsilon. I think they refer to alpha as the balancing parameter in line 238. It will help to name the parameter explicitly in that line. It will be very useful to give details on the regularization approach and how it fits with the logistic regression framework. What values of alpha did the regularization yield?

Following the reviewer's comment, we added a formal description of the logistic regression and regularization framework to the Supplement, p. 1. We also explained the choice of alpha and tested its robustness (Figure S1b), please see also answer to comment 3 of Reviewer 1.

14. Could the authors not have use the cross validation framework to set the value of epsilon as well rather than fix it arbitrarily at 0.1?

We chose the value for epsilon, which controls the number of interactions that are oriented, so that approximately 1/3 of the interactions remain unoriented in the consensus network, as suggested in ref. 14. Following the reviewer's comment as well as Reviewer 1 comment 3 above, we show that the observed enrichment of unoriented interactions with intra-complex interactions is robust to the value of epsilon used in a wide range from 0.05 to 0.15. We added this information to the Methods and Figure S1c.

Other Concerns:

1. Please use mathematical notation for the p-values, e.g., in line 151, the p-value as written seems to be “e to the power -12” rather than 10^{-12} .

Fixed.

2. The notation to describe the diffusion process can be improved. In line 216, the authors have not defined the set Priors. It will be so much cleaner if they first defined this set and then explained the diffusion process. The equations will look nicer if the authors use symbols instead of words, e.g., $w(u,v)$ instead of $\text{weight}(u,v)$. The weight term with a caret above it looks inelegant. Similarly, use P for the set of prior proteins. It will also help to explain in what sense these proteins are priors.

Fixed.

3. In lines 217-218, citation 34 is a paper by Yosef et al. Moreover, is there a need to cite a paper here unless the authors are using something other than standard power iteration?

We apologize for using a wrong reference here. We updated the reference (now ref. 41) to a recent review we have written on network diffusion. This is indeed standard computation, nevertheless we explained it per the request of Reviewer 3 (comment 7 below).

4. In lines 220-226, what does “computing a propagation” mean? That phrase is unclear. Perhaps the authors mean to say “We apply the diffusion procedure once by setting the causal proteins to be the set P of prior proteins. We use $F^C(u)$ to denote the score computed for each node u .” Later, they can say “We apply the diffusion procedure a second time after setting the set P to contain the effect proteins. We use $F^E(u)$ to denote the score computed for each node u .”

Fixed. We removed the reference to propagation to avoid confusion. Indeed it is the same as diffusion.

5. For the drug response data, did CMAP provide the drug target information as well?

No, the drug targets were extracted and assembled from DrugBank, DCDB, and KEGG DRUG databases. We apologize for the lack in details and we added them to the text (p. 13).

6. In line 442, the spelling of “parentheses” is wrong.

Fixed.

7. The 3D bars in Figure 4b and in Figure S3 seem unnecessary. 2D bars should be sufficient.

Fixed.

8. The authors say that the “unoriented” network is the one with all edge directions “flipped” compared to oriented network (lines 146-148, 154-156), which makes me think that the edge directions are simply reversed. However, the icon below the first bar in Figure 5a displays what appears to be mostly an undirected network as “unoriented network.” Is the icon incorrect? The

icons also include some colored shapes that the authors do not explain. The authors should consider simply deleting this icon.

We apologize for the confusion – the unoriented network is a network with no directions. When we flip (reverse) directions it is always with respect to the oriented network. We clarified the text accordingly and deleted the confusing icons (p. 9 and Figure 5).

9. It is unclear how the authors compute "orientation_confidence" column in Supplementary Information file. Are they the number of "votes for each direction from the five source-specific directed networks" (lines 139–140)? However, some edges have scores of 1.5. Moreover, won't the number of votes change based on the value of epsilon? I assume they used 0.1 here.

We revised the text to explain that the confidence is computed as the number of guiding sources supporting the inferred orientation divided by the number of guiding sources supporting the opposite orientation (p. 8). Indeed the confidence is affected by epsilon which is set to 0.1 throughout. We explain the choice of epsilon in response to comment 14 and to comment 3 of Reviewer 1.

Reviewer 3

I feel that the authors could perhaps make clearer what the main message of this paper is. The main finding of this work appears to be that an oriented PPI network is better than an unoriented one at predicting drug targets and cancer driver genes. This is, in fact, one of the results of the paper – in lines 142-175 the authors show that a very simple ranking method (based on proximity) gives better results at predicting drug targets and cancer driver genes when using an oriented PPI (oriented using the proposed method) than an un-oriented one. However, the title and the abstract could be interpreted as the authors proposing a method for improving the state-of-the-art prediction of drug targets and cancer driver genes – this is not the case here, as in order to show that, the authors would need to compare performances against state-of-the-art methods for these problems.

At the same time, most of the paper is about showing that the proposed method for orienting edges provides a better orientation than existing methods. Should this be the main message of this paper?

The main message of the paper is two-fold: First, the development of an orientation method for the human network that outperforms the state-of-the-art. Second, demonstrating its utility in gene prioritization with application to elucidating drug targets and cancer driver genes. We revised the title and abstract as suggested.

Many details are missing; I don't think we would be able to implement the algorithms described here. It would be useful to provide the datasets and the code which was run in the experiments.

Indeed as also requested by reviewer 1, We now published the code as an open source: <https://github.com/danasilv/Diffuse2Direct>

More specific comments:

(1) When reading the introduction, it feels like the authors are suggesting that the entire PPI network should be oriented. While the authors do refer to signalling circuitry, I found sentences like the one on lines 33-35 a bit misleading (“Nevertheless, to date, direction information is available for only a small percentage of the interactions. For example, KEGG contains 5,769 directed interactions in human out of the current 311,962 interactions present in BioGRID”). This issue becomes then clear later (lines 72-74), but I believe it would be easier to have a clearer overview in the introduction (and possibly in the abstract).

We revised the text accordingly (p. 2).

(2) The result on page 4, figure 2, is very good. But I have a couple of points:

a) There is only one short paragraph describing the experiment, at the beginning of page 4 with almost no details. Could the authors provide more details? Which datasets were used? How were parameters tuned? Etc.

We have revised and extended this result (p. 4 and Figure 2), see also answer to comment 7 of Reviewer 2 above.

b) More importantly, it would be interesting to understand whether such a good recovery of directionality for this pathways is an isolated case or whether it works in general. Would it be possible to quantify the method's performance on the entire KEGG? This would be much more convincing.

We thank the reviewer for this important suggestion. We repeated the analysis for the 5 largest pathways from KEGG (Mucin type O-glycan biosynthesis with 348 irreversible interactions, Glycosphingolipid biosynthesis with 330, Metabolism of xenobiotics by cytochrome P450 with 329, Steroid hormone biosynthesis with 224, and Fatty acid degradation with 212), predicting an edge (u,v) to be directed from u to v if its D2D score is greater than 1, to be directed from v to u if its D2D score is smaller than 1, and to be an undirected edge if its D2D score is equal to 1. Reassuringly, we found that our predictions for 88% of the edges agreed with the KEGG annotations. We added these results to Figure 2b as well as a D2D score distribution for the different cases to Figure 2c, showing that our method can differentiate well between true, undirected and false direction annotations.

(3) About the second results, “drug response to orient a human PPI network”, starting on line 81.

a) In Table S1, the authors should expand what they mean by “Instances (50% correct orientations + 50% incorrect orientations)” – it is not easy to follow, and it becomes clear only when one connects it to the fact that there are 2 entries for each connection in the output of their method.

The text has been revised (p. 16 and SI p. 3 below Table S1), see also answer to comment 9b of Reviewer 2 above.

b) In the same S1 table, I believe that the last column (“Benchmark orientation”) refers to the DADA approach, but it is not clear.

Thanks, we explained that this is a topology-based orientation (similar in spirit to the DADA degree correction, see p. 15).

c) Figure 3: why some of the green curves don't reach 0.5?

In these cases the curves reached a recall of 1 with precision above 0.5.

(4) About the third result, “using gene pairs derived from genomic mutation data and the resulting expression changes” to orient a human PPI network, starting on line 120: it would be useful to compare the result with the methods of Vinayagam and SHORTEST, showing AUPR curves – just as it was done for the previous (second) set of results.

While we focus the comparisons on the drug response dataset, we now added the requested information. We note that the Vinayagam guiding pairs do not depend on the analyzed dataset, hence the orientation solution will not change and is outperformed by D2D. We added the AUPR results yielded by Vinayagam to Table S1 to reflect that. The method SHORTEST is very limited in its coverage, thus its AUPR is much poorer than either Vinayagam or D2D (this limitation is discussed on page 6 of the manuscript).

(5) About the fourth result, starting on line 142.

I could not understand what the authors mean here: “To this end, we flipped all directions in the network and computed a network diffusion score using the differentially expressed genes induced by the drug as a prior set.”

In the same way, on line 154, this sentence was not clear: “As before we flipped the network, and computed the network diffusion scores for each patient separately, diffusing from the patient differentially expressed genes.”

My understanding is that the authors are testing whether an oriented network is better than an unoriented one at predicting drug targets and cancer driver genes, but it would be easier for the reader if they expanded their explanation.

We apologize for the confusion. When we flip (reverse) directions it is always with respect to the oriented network. We flip the directions when we diffuse from the differentially expressed genes because we would like to diffuse “upwards” in the network, thus finding the mutations responsible for the changes in gene expression. We have expanded the pertaining text to clarify these issues (p. 9).

(6) In general, the output of the proposed method is a partially oriented graph, where some edges are directed, and some undirected. The authors tested and quantified the correctness of the directed edges. I feel that it would make sense to also test and quantify the correctness of the edges that are left undirected by the algorithm. Test sets for this could be obtained, for example, from known protein complexes.

We thank the reviewer for this important suggestion! We added a paragraph evaluating the consensus network (p. 8). We found that the edges left unoriented in the network (about 31% of the edges with orientation confidence less than 2) are highly enriched with known complexes from the CORUM database with a hypergeometric p-value of $1.7e-103$. For comparison, when applying the same test to the highly confident interactions (confidence of 5), we obtained an insignificant p-value of 0.99.

(7) It would be important to have a more detailed explanation of the diffusion method – explained beginning in line 210. As it stands, I don't feel the paper is self-contained – it refers to explanations from the Cowen et al paper which I think would be good to include here. Also, in the explanation on page 9, it was unclear to me: how many iterations is the algorithm run for? What is the “prior set”? It would also be good to include some intuition for the method.

We added the required information to make the description self contained and provide more intuition about the diffusion process. The diffusion algorithm is guaranteed to converge in a connected network, where convergence is achieved when the square root of the summed absolute changes for an iteration is below β , where we set β to be 10^{-5} (p. 12).

(8) In general, 3-fold cross validation is not ideal – it is not too far from just using training and testing, and no validation. Why did the authors not use the more standard 10-fold cross validation? I suspect that the numerosity of the dataset did not allow for it, but I feel it would be useful to discuss this point.

We chose 3-fold for efficiency. As requested, we repeated the analyses with 5 and 10 folds, also requested by reviewer 1 comment 9c, and found that the results are consistent with a slight increase in performance as the number of folds increase. We added the analysis to the Supplementary Information, Figure S1a.

(9) Lines 235-240 were not clear to me. Many details are missing. (Again, the formulas for the logistic regression classifier with L1 regularization (possibly in the supp mat) would make the paper more self-contained). Which parameters are learned through a 3-fold cross validation? Does the “balancing parameter” on line 238 refer to the alpha parameter on line 214, or does it have to do with the L1 regularization? Where does value of 0.1 for epsilon come from?

We added the requested information to the methods. We clarified that the balancing parameter is related to the regularization.

We chose the value for epsilon, which controls the number of interactions that are oriented, so that approximately 1/3 of the interactions remain unoriented in the consensus network, as suggested in ref. 14. Following the reviewer's comment as well as Reviewer 1 comment 3 above, we show that the observed enrichment of unoriented interactions with intra-complex interactions is robust to the value of epsilon used in a wide range from 0.05 to 0.15. We added this information to the Methods and Figure S1c.

(10) The sentence “we filtered the test sets by random down-sampling so that similar numbers of interactions from high to low degree and from low to high degree proteins remain.” was not very clear to me. Could you expand? Could you describe the algorithm in detail? (it could go in the supplementary material)

We extended the text to clarify the down-sampling (p. 16). Please also see answer to comment 4 of Reviewer 1 above.

MINOR COMMENTS

Fig 1a, green and blue circle could be used in the different layers

Fixed.

Line 218, the reference should probably be 37

Fixed. The ref. is now 41.

Ranking of driver genes per left-out cancer using a consensus network oriented using the remaining guiding sources. The genes in each network are ranked by their proximity to the differentially expressed genes of the cancer patients. Showing here the precision recall curves up to 0.1 for all control lists across the four cancers in the paper.

Reviewers' comments:

Reviewer #1 (Remarks to the Author):

I thank the authors for addressing my major concerns to a great extent. I list few remaining issues that in my opinion, need further attention below:

1. The VEGF case study is extremely encouraging but it might be causing over-excitement due to the nature of this pathway: All the interactions are from input toward outputs, which are picked up by the diffusion process as the score is spread from VEGF to the other nodes inversely correlated to the distance from the VEGF. Accordingly, all the interactions that are from a node with a lower DFS (depth first search) visit timestamp to another node with higher or equal DFS visit timestamp are incorrectly assigned by the proposed method. I suspect this reflects the nature of the information flow in the KEGG pathways used in Fig 2 and tells little about the power of the D2D in general. Could the authors refute this claim, e.g., by assigning directions with the simple DFS visit timestamp strategy I mentioned above and showing that it performs significantly worse than D2D?

2. The readability of the text has substantially improved after the authors' revision. Still, having different inputs & outputs (cause/effect nodes used for generating directed edge score via diffusion process) as well as training & test sets (used for benchmarking logistic regression model on different data sets via cross validation) makes the text hard to follow. The authors first employ diffusion on drug target and expression data to generate scores to be fed to classifier using 5 benchmarking data sets to validate the networks, then employ diffusion on TGCA mutation and COSMIC differential expression data to generate scores to be fed to classifier (validated on the same benchmarking sets?), which is later used to predict proximity to driver genes. I would suggest to add a table containing the information on input & output data as well as training & test sets for each logistic regression based model they trained as well as each network they generated in the main text along with the size of these data sets (cause & effect nodes, positive & negative directed interactions used in the training step, directed interactions in the final predicted network).

3. I still feel that the random down-sampling procedure is unclear. For instance, how low and high degree nodes were defined?

4. Despite the change from the previous revision, benchmark label in Fig 3. continues to be ambiguous for me, it could maybe be replaced by "only topology".

5. Briefly explaining the procedure used by Iskar et al. would help to understand how effect proteins are decided using drug response data.

Minor issues:

- cross-validation / cross validation inconsistently used
- in total*, *
- sub*s*tract, non-dr*o*ver (in Supplementary)

Reviewer #2 (Remarks to the Author):

The authors have revised the manuscript very substantially. They have addressed my queries very well. I like several of the new analyses they have performed, e.g., testing enrichment in complexes, checking predicted orientations against other datasets, and studying the features of drugs that contribute to the classifier. I noted that the correlations between drugs with non-zero coefficients and their feature are low (the significance may be high simply because of the large number of drugs). Nevertheless, the trends shed some light on what features drive the success of the logistic regression classifier.

I have only a few minor comments.

line 60: I suggest changing "Diffuse2Direct is available as an open source:" to "Diffuse2Direct is available as an open source package:"

line 119: The phrase "diffusion suggested in 23" looks strange, especially with the citation number appears as a superscript. Adding the author names before the number might be a good idea.

line 120: In "benchmark results in an average of 0.56", it will help to "area under the curve" after "average".

line 192: Please replace the "." in the p-value with a multiplication symbol (x).

line 203: Would "wrongly-oriented" or "flipped" be a better term than "unoriented" to describe the network where the authors flip the orientations of the edges?

line 242: "solution" should be in the plural.

Reviewer #3 (Remarks to the Author):

The authors have done a great amount of work and the manuscript has improved immensely. Also the images and supplementary information that have been added are extremely useful.

I still have a couple of points:

1) On Figure 2:

When I read the first version of the manuscript many details were missing, and I was under the impression that somehow the figure had been produced using the PPI network with the pathway inputs and outputs as the "causal" and "affected" proteins, and that the edge directions inferred from this diffusion matched the real direction in 48/52 cases. This would have been a very impressive result.

However, I understand now that the analysis shown in Figure 2 was not carried out using PPI networks, but using metabolic pathways. Pathways edges are inherently directed from input nodes to output nodes, therefore it is not surprising at all that the diffusion gradient will decrease along the pathway flow direction and most of the arrows will be oriented correctly. As a matter of fact, I believe it would be relatively easy to come up with a very simple algorithm that would traverse the graph to orient correctly the pathways and have even a better accuracy. This is because the diffusion-based algorithm can make mistakes in the presence of "shortcuts" between "causal" and "affected" proteins, as it is the case, for example, in Fig 2, on the Ca-PCK connection. Maybe the authors should mention this point, at least in the discussion.

2) On the title:

Although the title has been modified, I think it is still misleading. This paper presents a method for orienting a network. The part on elucidating drug targets and cancer driver genes seems to me to be really a smaller point, occupying just the last page of the paper. Also, the title in the Supplementary Material has not changed.

3) On the benchmark score names:

I believe that the description of the benchmark given in the main manuscript is still a bit misleading. The authors mention on lines 349 that DADA scores are calculated, and yet DADA is not referred to in Table S1. Moreover, as I understood, the benchmark orientation scores are a special case of the proposed algorithm with $\alpha = 1$. There are too many labels for this method and this makes it confusing.

Reviewer #1

1. The VEGF case study is extremely encouraging but it might be causing over-excitement due to the nature of this pathway: All the interactions are from input toward outputs, which are picked up by the diffusion process as the score is spread from VEGF to the other nodes inversely correlated to the distance from the VEGF. Accordingly, all the interactions that are from a node with a lower DFS (depth first search) visit timestamp to another node with higher or equal DFS visit timestamp are incorrectly assigned by the proposed method. I suspect this reflects the nature of the information flow in the KEGG pathways used in Fig 2 and tells little about the power of the D2D in general. Could the authors refute this claim, e.g., by assigning directions with the simple DFS visit timestamp strategy I mentioned above and showing that it performs significantly worse than D2D?

As requested by the reviewer, we implemented DFS search as well as an alternative BFS search and found that they perform significantly worse than D2D. While D2D achieves 92% accuracy in orientation prediction, DFS achieves only 57% accuracy and BFS achieves 81%. We added these results to the manuscript (pages 4-5). We observed that the diffusion based orientation benefits from diffusing a signal to multiple edges at the same time, free of DFS and BFS limitation of diffusing a signal one edge at a time, resulting in more accurate orientations even in this simple setting (where a pathway rather than a whole network is being oriented).

2. The readability of the text has substantially improved after the authors' revision. Still, having different inputs & outputs (cause/effect nodes used for generating directed edge score via diffusion process) as well as training & test sets (used for benchmarking logistic regression model on different data sets via cross validation) makes the text hard to follow. The authors first employ diffusion on drug target and expression data to generate scores to be fed to classifier using 5 benchmarking data sets to validate the networks, then employ diffusion on TGCA mutation and COSMIC differential expression data to generate scores to be fed to classifier (validated on the same benchmarking sets?), which is later used to predict proximity to driver genes. I would suggest to add a table containing the information on input & output data as well as training & test sets for each logistic regression based model they trained as well as each network they generated in the main text along with the size of these data sets (cause & effect nodes, positive & negative directed interactions used in the training step, directed interactions in the final predicted network).

We thank the reviewer for the great suggestion and added a table S2 (SI text, page 5) as requested to improve the readability of the manuscript.

3. I still feel that the random down-sampling procedure is unclear. For instance, how low and high degree nodes were defined?

We clarified the text (page 16) to explain that the goal is to have similar numbers of edges going from a node with a lower degree to a node with a higher degree and vice versa (from a node with a higher degree to a node with a lower degree).

4. Despite the change from the previous revision, benchmark label in Fig 3. continues to be ambiguous for me, it could maybe be replaced by "only topology".

Fixed to "topology only".

5. Briefly explaining the procedure used by Iskar et al. would help to understand how effect proteins are decided using drug response data.

For completeness, we added the required information to the supplementary material (SI text, page 2).

Minor issues:

- cross-validation / cross validation inconsistently used
- in total*, *
- sub*s*tract, non-dr*o*ver (in Supplementary)

All minor issues were fixed.

Reviewer #2

I have only a few minor comments.

line 60: I suggest changing "Diffuse2Direct is available as an open source:" to "Diffuse2Direct is available as an open source package:"

The code consists of a c++ part and a python part, hence it is not a package per se. We did fix the line following the reviewer suggestion to "diffuse2direct is available open source on github at:".

line 119: The phrase "diffusion suggested in 23" looks strange, especially with the citation number appears as a superscript. Adding the author names before the number might be a good idea.

Fixed.

line 120: In " benchmark results in an average of 0.56", it will help to "area under the curve" after "average".

Fixed.

line 192: Please replace the "." in the p-value with a multiplication symbol (x).

Fixed.

line 203: Would "wrongly-oriented" or "flipped" be a better term than "unoriented" to describe the network where the authors flip the orientations of the edges?

That unfortunately would not be the correct terminology. The flipping refers to the fact that we assume that in an oriented network edges should be directed from a drug target to the drug's differentially expressed genes. Thus, to find a target we diffuse from those differentially expressed genes by flipping all edge directions first. We compare the result to that obtained when the network is not oriented to begin with (so no flipping in this case). We clarified the text accordingly (page 9).

line 242: "solution" should be in the plural.

Fixed.

Reviewer #3

1) On Figure 2:

When I read the first version of the manuscript many details were missing, and I was under the impression that somehow the figure had been produced using the PPI network with the pathway inputs and outputs as the "causal" and "affected" proteins, and that the edge directions inferred from this diffusion matched the real direction in 48/52 cases. This would have been a very impressive result.

However, I understand now that the analysis shown in Figure 2 was not carried out using PPI networks, but using metabolic pathways. Pathways edges are inherently directed from input nodes to output nodes, therefore it is not surprising at all that the diffusion gradient will decrease along the pathway flow direction and most of the arrows will be oriented correctly. As a matter of fact, I believe it would be relatively easy to come up with a very simple algorithm that would traverse the graph to orient correctly the pathways and have even a better accuracy. This is because the diffusion-based algorithm can make mistakes in the presence of "shortcuts" between "causal" and "affected" proteins, as it is the case, for example, in Fig 2, on the Ca-PCK connection. Maybe the authors should mention this point, at least in the discussion.

We thank the reviewer for their suggestion. Per also the request of Reviewer 1, we implemented simple orientation approaches to this specific setting and demonstrated the advantage of our new D2D method, please see answer to first question of reviewer 1.

2) On the title:

Although the title has been modified, I think it is still misleading. This paper presents a method for orienting a network. The part on

elucidating drug targets and cancer driver genes seems to me to be really a smaller point, occupying just the last page of the paper. Also, the title in the Supplementary Material has not changed.

Our paper is the first to show the utility of network orientation in predicting biomedical-related sets of proteins. We have shown that using the oriented network consistently yields better predictions across parameter values choices and validations sources, hence we prefer to keep our current title. We fixed the supplementary material title as requested.

3) On the benchmark score names:

I believe that the description of the benchmark given in the main manuscript is still a bit misleading. The authors mention on lines 349 that DADA scores are calculated, and yet DADA is not referred to in Table S1. Moreover, as I understood, the benchmark orientation scores are a special case of the proposed algorithm with $\alpha = 1$. There are too many labels for this method and this makes it confusing.

We thank the reviewer for highlighting this point. We corrected the text to clarify that the scores are "DADA-inspired" scores and not DADA scores, and now refer to the method by the same name in all results ("topology only" as asked by Reviewer 1, in Figure 3 and Table S1).

REVIEWERS' COMMENTS:

Reviewer #1 (Remarks to the Author):

I thank the authors for having addressed my earlier comments and recommend the acceptance of the manuscript for publication.